# Propionyl-CoA carboxylase subunit B regulates anti-tumor T cells in a pancreatic cancer mouse model

Han V Han[1,2], Richard Efem[1], Barbara Rosati[1], Kevin Lu[3], Sara Maimouni[3], Ya-Ping Jiang[1], Valeria Montoya[4,5], Ando Van Der Velden[5], Wei-Xing Zong[3], Richard Z Lin[1,6]*

[1]Department of Physiology and Biophysics, Stony Brook University, Stony Brook, New York, United States; [2]Department of Biomedical Engineering, Stony Brook University, Stony Brook, New York, United States; [3]Department of Chemical Biology, Ernest Mario School of Pharmacy, Rutgers-The State University of New Jersey, Piscataway, United States; [4]Department of Microbiology and Immunology, Renaissance School of Medicine at Stony Brook University, Stony Brook, New York, United States; [5]Center for Infectious Diseases, Renaissance School of Medicine at Stony Brook University, Stony Brook, New York, United States; [6]Northport Veteran Affair Medical Center, Northport, New York, United States

## eLife assessment

The significance of the findings is **valuable**, with implications for immunotherapy design in pancreatic ductal adenocarcinoma. The evidence was considered **incomplete** and partially supportive of the major claims.

*For correspondence: richard.lin@stonybrook.edu

**Abstract** Most human pancreatic ductal adenocarcinoma (PDAC) are not infiltrated with cytotoxic T cells and are highly resistant to immunotherapy. Over 90% of PDAC have oncogenic KRAS mutations, and phosphoinositide 3-kinases (PI3Ks) are direct effectors of KRAS. Our previous study demonstrated that ablation of *Pik3ca* in KPC (*Kras*[G12D]; *Trp53*[R172H]; *Pdx1-Cre*) pancreatic cancer cells induced host T cells to infiltrate and completely eliminate the tumors in a syngeneic orthotopic implantation mouse model. Now, we show that implantation of *Pik3ca*[−/−] KPC (named αKO) cancer cells induces clonal enrichment of cytotoxic T cells infiltrating the pancreatic tumors. To identify potential molecules that can regulate the activity of these anti-tumor T cells, we conducted an in vivo genome-wide gene-deletion screen using αKO cells implanted in the mouse pancreas. The result shows that deletion of propionyl-CoA carboxylase subunit B gene (*Pccb*) in αKO cells (named p-αKO) leads to immune evasion, tumor progression, and death of host mice. Surprisingly, p-αKO tumors are still infiltrated with clonally enriched CD8[+] T cells but they are inactive against tumor cells. However, blockade of PD-L1/PD1 interaction reactivated these clonally enriched T cells infiltrating p-αKO tumors, leading to slower tumor progression and improve survival of host mice. These results indicate that *Pccb* can modulate the activity of cytotoxic T cells infiltrating some pancreatic cancers and this understanding may lead to improvement in immunotherapy for this difficult-to-treat cancer.

## Introduction

Pancreatic ductal adenocarcinoma (PDAC) is one of the most aggressive and lethal cancers, and it is expected to be the second leading cause of cancer death by 2030, with a 5-year survival rate of 12% (*Siegel et al., 2023*). Most PDACs are classified as 'cold' tumors without infiltrating T cells and are resistant to currently available immunotherapy with checkpoint inhibitors (*Bian and Almhanna, 2021*; *Mucileanu et al., 2021*; *Balsano et al., 2023*). To convert PDAC into 'hot' tumors, researchers have explored multiple strategies to promote immune responsiveness of PDAC (*Fan et al., 2020*; *Johnson et al., 2017*; *Halbrook et al., 2023*). These approaches encompass targeting immune checkpoint inhibitors, including PD1, CTLA4, LAG3, and TIGIT (*Bian and Almhanna, 2021*; *Liu et al., 2022*; *Li et al., 2021*; *Freed-Pastor et al., 2021*; *Gulhati et al., 2023*), suppressing immunosuppressive cells such as myeloid-derived suppressor cells and/or tumor-associated macrophages (*Panni et al., 2019*; *Vonderheide, 2020*; *Zhu et al., 2017*; *Wu et al., 2022*), enhancing antigen processing and epitope presentation by promoting MHC class I/II cell surface expression (*Sivaram et al., 2019*; *Baleeiro et al., 2022*; *Yamamoto et al., 2020*) and reinstating the expansion and functionality of dendritic cells (*Hegde et al., 2020*), targeting key factors implicated in stromal fibrosis to facilitate the infiltration of T cells (*Jiang et al., 2016*; *Özdemir et al., 2014*; *Shi et al., 2014*), and depleting immunosuppressive signals (cytokines and chemokines) in the tumor microenvironment (TME) that hinder the entry and function of T cells (*Popovic et al., 2018*; *Mace et al., 2018*; *Dey et al., 2020*). However, a previous study showed that a minority of PDAC patients have tumors that already exhibit higher levels of infiltrating cytotoxic T cells and this observation is associated with longer survival of these patients (*Balachandran et al., 2017*).

One of the most commonly dysregulated pathways in PDAC is oncogenic KRAS and G12D is the most common KRAS mutation (46%) in PDAC (*Halbrook et al., 2023*; *Moore et al., 2020*). Phosphatidylinositol 3-kinases (PI3Ks) are direct effectors of KRAS. There are four PI3K catalytic isoforms and our group have previously shown that PIK3CA isoform plays a critical role in the initiation of PDAC (*Wu et al., 2014*) and mediates immune evasion once the tumors are formed (*Sivaram et al., 2019*). Our group reported that KRAS signaling through PIK3CA can reduce the expression of MHC class I molecules, thus reducing antigen recognition and T-cell infiltration (*Sivaram et al., 2019*). We showed that genetic ablation of PIK3CA in KPC ($Kras^{G12D}$;$Trp53^{R172H}$;$Pdx1$-$Cre$) pancreatic tumor cells led to T-cell recognition and complete elimination by the host immune system in a syngeneic orthotopic implantation mouse model (*Sivaram et al., 2019*). Using this well-characterized syngeneic orthotopic implantation mouse model with the αKO ($Kras^{G12D}$;$Trp53^{R172H}$;$Pdx1$-$Cre$;$Pik3ca^{-/-}$) cell line and single-cell sequencing analysis, we found clonal enrichment of cytotoxic T cells infiltrating αKO tumors. To investigate potential molecules within αKO cells that affect the activity of host T cells, we performed an in vivo genome-wide CRISPR gene-deletion screen using αKO cells and found that deletion of propionyl-CoA carboxylase subunit B (PCCB) could reverse the immune-elimination phenotype. We further discovered that T cells infiltrating the p-αKO ($Kras^{G12D}$;$Trp53^{R172H}$;$Pdx1$-$Cre$;$Pik3ca^{-/-}$;$Pccb^{-/-}$) tumors were suppressed and could not eliminate the tumor cells. However, these exhausted anti-tumor T cells could be reactivated by neutralizing the PD-L1/PD1 axis, restoring the ability of these immune cells to eliminate the pancreatic tumor.

## Results

### Clonal enrichment of cytotoxic T cells infiltrating the pancreatic αKO tumors

In our previous study, we demonstrated that αKO tumors, when implanted in immunocompetent C57BL/6 (B6) mice, are recognized and eliminated by the host immune system, with cytotoxic T cells playing a pivotal role in this process (*Sivaram et al., 2019*). To investigate the status of the cytotoxic T cells infiltrating to the TME, single-cell RNA sequencing (scRNA-seq) was conducted in combination with concurrent T-cell receptor (TCR) repertoire sequencing. On day #12 post-implantation in two B6 mice, both αKO tumors were harvested and dissociated to generate single-cell suspensions. CD8[+] T cells were isolated using microbead preparations, and both the transcriptional expression and TCR repertoire profiles were obtained using 10× genomic protocols (*Figure 1A*). Following stringent quality control filters (outlined in the Methods section), we successfully obtained gene expression profiles and TCR sequences from the same cytotoxic T cells.

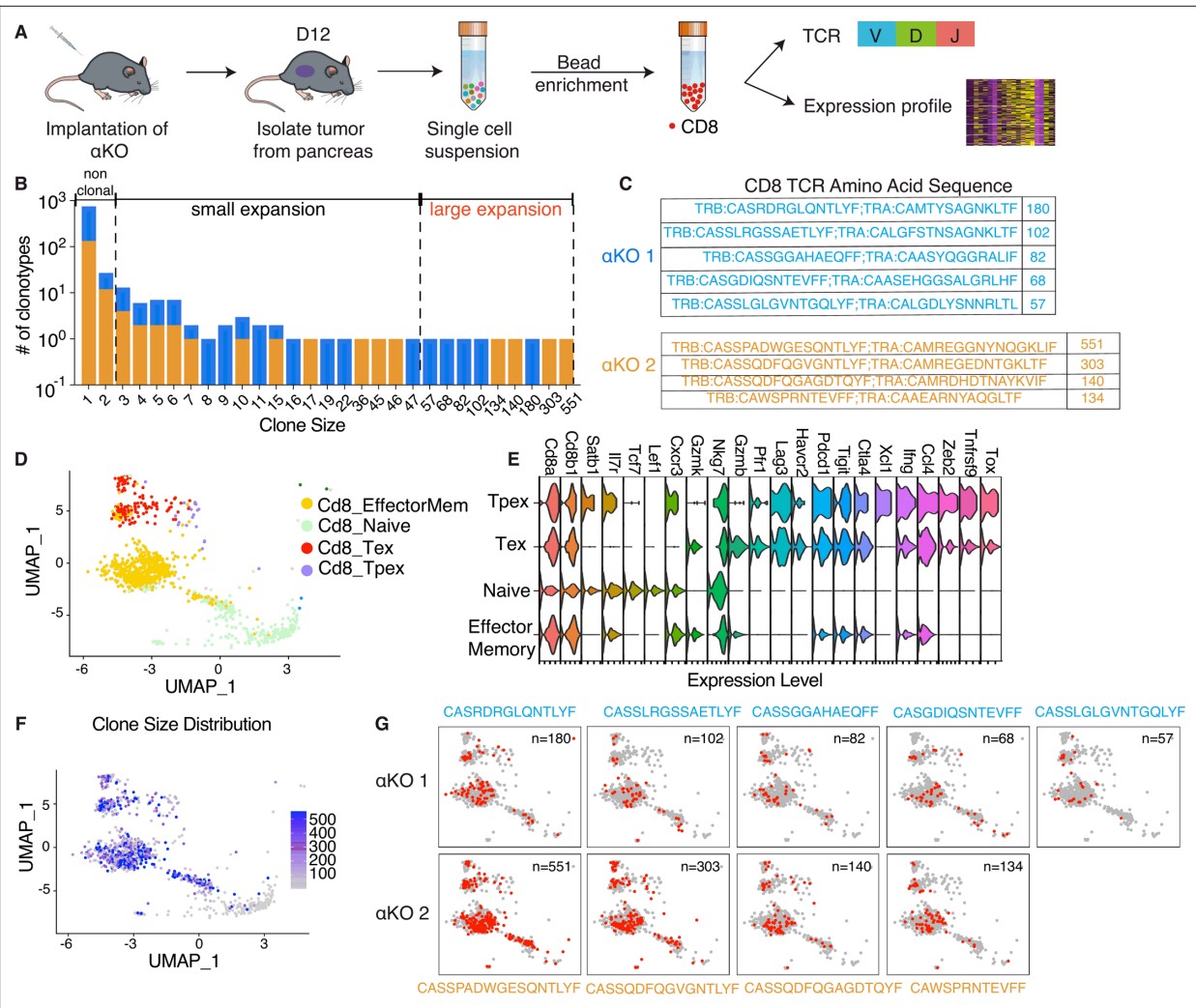

**Figure 1.** Clonal enrichment of cytotoxic T cells infiltrating the αKO pancreatic tumors. (**A**) Experimental design schema for single-cell RNA sequencing of infiltrating CD8+ T cells in αKO tumors. Cells were implanted in the pancreas of two B6 mice and tumors were harvested at 12 days post-implantation. Single-cell suspensions were enriched with CD8+ beads followed by concurrent single-cell transcriptional and T-cell receptor (TCR) profiling using 10× Genomics protocols. In total, 5075 CD8+ cells were identified after quality control. (**B**) Size distribution for CD8+ T-cell clonal types in αKO tumors. Non-clonal: $N \leq 2$ cells; small expansion: 3 cells $\leq N <$ 50 cells; large expansion: $N \geq$ 50 cells. (**C**) Amino acid sequences of the CDR3 region for large expansion clonotypes. αKO1 tumor has five and αKO2 tumor has four large expansion clonotypes (full list shown in **Supplementary file 1**). (**D**) Uniform manifold approximation and projection (UMAP) of all CD8+ T-cell clonotypes. Principle component analysis of gene expression profiles generated four-cell clusters annotated based by ProjectTILs. EffectorMem: effector memory; Tex: exhausted; Tpex: precursor exhausted; Naive: naive like or central memory. Each dot corresponds to a single cell. (**E**) Major markers used for cell type annotation by Projectils. (**F**) Clonal size distribution mapped to the gene expression UMAP showing that the largest clonotypes are mostly effector memory T cells. (**G**) The gene expression profile distribution of individual large expansion clonotypes are shown. The majority of cells for each clone mapped to the effector memory T-cell functional group.

The online version of this article includes the following source data for figure 1:

**Source data 1.** Clone size counts containing data for **Figure 1B**.

**Source data 2.** TCR clonotype CDR3 sequences and counts from two αKO tumors for **Figure 1C**.

The diverse expression of TCR repertoires is crucial for effective adaptive immune responses. Sequencing the TCR repertoire allows us to identify clonotypic diversity and detect clonal enrichment (**Tu et al., 2019**). Analysis of the CD8+ T cells' clonal size distribution revealed clonal enrichment in both αKO tumors, suggesting T cells in both tumors have been activated in response to tumor antigen (**Figure 1B**). A clonotype is defined as three or more cells with identical TCR sequences. A clonal size of 3–49 is defined as small expansion, and a clonal size of 50 or more is designated as large expansion

(*Figure 1B*). *Figure 1C* displays the TCR sequences for CD8$^+$ clones with large expansions (clonal size ≥50) in both tumors.

We next analyzed the clonality of these cells with respect to their transcriptomes. The gene expression profiles were visualized using the uniform manifold approximation and projection (UMAP) (*Figure 1D*). To achieve this, the expression data underwent unsupervised clustering by Seurat. Cell type annotations were determined using ProjectTILs, a computational method that annotates cell types by projecting scRNA-seq data onto reference profiles constructed from well-established canonical cell markers and clinical data (*Andreatta et al., 2021*). The Violin Plot in *Figure 1E* illustrates major markers used for cell type annotation by ProjectTILs. Based on the annotation, clusters are annotated as effector memory CD8$^+$ T cells (Cd8_EffectorMem), naive-like CD8$^+$ T cells or central memory T cells (Cd8_NaiveLike), exhausted CD8$^+$ T cells (Cd8_Tex), and precursor exhausted CD8$^+$ T cells (Cd8_Tpex). We next mapped the TCR profiling datasets to the single-cell gene expression datasets and then color coded by the size of each clonotype (*Figure 1F*). The degree of expansion

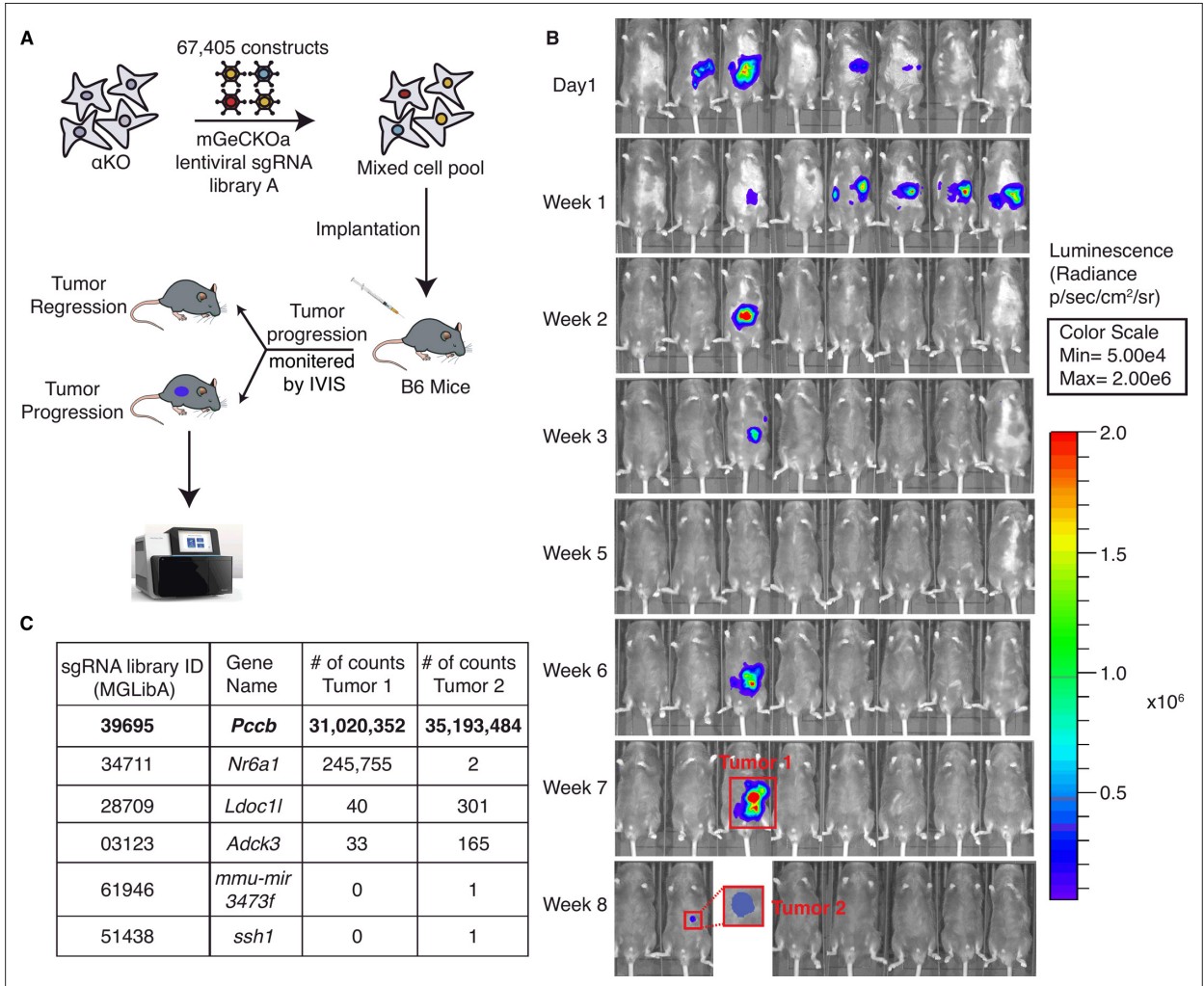

**Figure 2.** A genome-wide CRISPR gene-deletion screen identifies *Pccb* contributing to PIK3CA-mediated pancreatic tumor immune evasion. (**A**) A schema for the genetic screen. Mouse genome-scale CRISPR/Cas9 knockout library (mGeCKO v2A) containing three sgRNA each for 20,611 genes was transduced into *Kras*$^{G12D/+}$;*Tp53*$^{R172H/+}$;*Pik3ca*$^{-/-}$ (αKO) cells. Pooled cells were implanted in the pancreas of eight C57BL/6J mice. (**B**) IVIS imaging was performed to monitor tumor progression in these mice. In two animals, tumors were still present at the end of the screen (annotated as tumors 1 and 2). (**C**) Harvested DNAs prepared from the tumors were subjected to next-generation sequencing (NGS) to identify enriched sgRNAs barcodes. Sequences corresponding to only six genes in the mGeCKO v2A library have detectable NGS counts as shown in the table. In both tumors, sgRNA sequences mapping to the *Pccb* gene are most numerous (full list shown in *Supplementary file 2*).

The online version of this article includes the following source data for figure 2:

**Source data 1.** sgRNA counts containing data for *Figure 2C*.

observed among the clonotypes associated strongly with the phenotypic clusters of T cells, as large clones are mapped to the Cd8_EffectorMem and Cd8_Tex, while non-clonal T cells mapped to the Naive/centra memory T-cell group (*Figure 1F*). *Figure 1G* shows the single-cell gene profile distribution of large expansion (clonal size ≥50) clonotype from both tumors. Taken together, these results show that there is a massive expansion of Cd8_EffectorMem clones and these activated anti-tumor T cells are likely are responsible for the immune-mediated elimination of αKO pancreatic tumors.

## A genome-wide CRISPR gene-deletion screen to identify molecules contributing to Pik3ca-mediated pancreatic tumor immune evasion

To identify potential molecules in tumor cells that can regulate the activity of these anti-tumor clonal T cells, we conducted an in vivo genome-wide gene-deletion screen using αKO cells implanted in the mouse pancreas. A schema for the CRISPR knockout screen is shown in *Figure 2A*. Briefly, αKO cells were first infected with the genome-scale CRISPR knockout (GeCKO v2) pooled lentiviral library A, which targets the complete mouse genome of 20,611 genes with three sgRNAs for each gene, as well as four sgRNAs for each of the 1175 miRNAs and 1000 control sgRNAs. Equal portion of the mixed cell pool was implanted into the pancreas of eight immunocompetent B6 mice and tumor progression was monitored longitudinally by an IVIS Lumina III in vivo imaging system (*Figure 2B*). When implanted in B6 mice, αKO tumors completely regressed. Therefore, cells with genetically deleted molecules that can reverse the immune recognition phenotype should lead to tumor progression. Indeed, two out of eight mice 8 weeks after cell implantation exhibited tumor progression, which were labeled as tumors #1 and #2 (*Figure 2B*). Tumors were harvested from both mice for further analysis. The remaining mice (six out eight) exhibited complete tumor regression consistent with the phenotype we previously observed with αKO cells (*Figure 2B*). Genomic DNA was prepared from both tumors and subjected to next-generation sequencing (NGS) to detect the presence of incorporated sgRNA sequences. Sequenced data were analyzed using model-based analysis of genome-wide CRISPR/Cas9 knockout (MAGeCK), and the counts for each sgRNA sequence are shown in *Figure 2C* (full list shown in *Supplementary file 2*). Somewhat surprisingly, four common sgRNA sequences were identified in both tumors. The four targeted genes are *Pccb* (propionyl-CoA carboxylase), *Nr6a1* (nuclear receptor subfamily 6 group A member 1), *Ldoc1l* (leucine-zipper protein 1), and *Adck3* (aarF-domain-containing kinase 3). Tumor #2 has two additional sgRNA sequences targeting two other genes that were not present in tumor #1. Despite the fact that tumors #1 and #2 grew at different rates and were harvested at different time points, weeks 7 and 13, respectively, the detected sgRNAs were highly similar, strongly suggesting that the identified genes play critical roles in *Pik3ca*-mediated immune evasion.

In both tumors, *Pccb* sgRNA sequence MGLibA ID 39695 was detected at many orders of magnitude higher than other three sgRNA sequences (*Figure 2C*). Propionyl-CoA carboxylase (PCC) has two subunits PCCA (propionyl-CoA carboxylase subunit A) and PCCB and catalyzes the carboxylation of propionyl-CoA to methylmalonyl-CoA, which is then converted to succinyl-CoA that enters the tricarboxylic acid cycle (TCA). PCC dysfunction leads to a rare genetic disease called propionic acidemia, which has been extensively studied (*Wongkittichote et al., 2017*; *Shchelochkov et al., 1993*; *Grünert et al., 2013*). However, PCCB's role in cancer and whether it plays a role in the tumor immune response is completely unknown.

## Ablation of PCCB reverses the immune recognition of αKO cells

To validate the genetic screen result, we next genetically ablated *Pccb* in αKO cells using CRISPR/Cas9 techniques and generated the PCCB-null p-αKO cell line. PCCB ablation was confirmed by western blotting (*Figure 3A*). We then orthotopically implanted p-αKO cells in the head of pancreas of B6 mice and monitored tumor growth by IVIS imaging (*Figure 3B*). Most of the mice exhibited tumor formation within 1 week. In some of the host animals, tumor regressed when they were reimaged at week #2. However, in all the animals, pancreatic tumors eventually progressed and 9 out of 12 mice died by week #8 when the experiment was stopped (*Figure 3B, C*). Postmortem examination showed that the three surviving mice had large pancreatic tumors that were also detectable by IVIS imaging (*Figure 3B*). In comparison, while 100% of mice implanted with KPC cells died by week #3, all mice implanted with αKO cells remained tumor free by week #8 (*Figure 3C, D*). These results indicate that PCCB loss can reverse immune recognition of αKO cells.

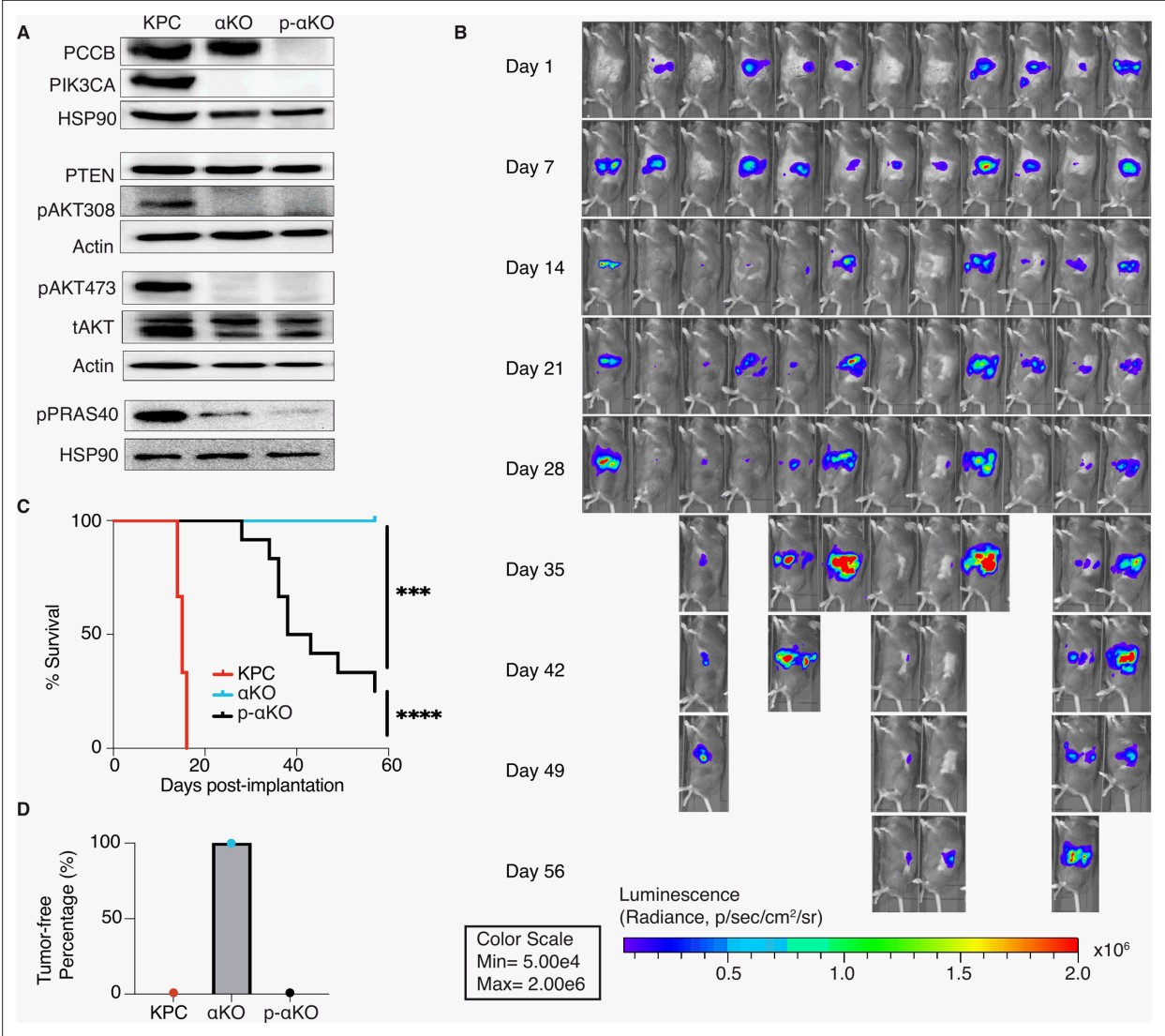

**Figure 3.** Ablation of propionyl-CoA carboxylase subunit B (PCCB) reverses the immune recognition of αKO cells. (**A**) *Pccb* was deleted in αKO cells by CRISPR/Cas9 to generate p-αKO cells. Western blotting confirmed PCCB and PIK3CA loss in p-αKO cells. Additionally, western blotting was performed for PTEN, phospho-Thr308 AKT, phospho-Ser473 AKT, and phospho-PRAS40, with Actin and HSP90 used as loading controls. (**B**) Orthotopic implantation of p-αKO tumors in the pancreas of C57BL/6J mice was monitored by IVIS imaging of the luciferase signal weekly for 8 weeks. (**C**) Kaplan–Meier survival curves for mice implanted with KPC, αKO, and p-αKO cells. Median survival KPC: 15.5 days ($N = 8$); αKO, all alive ($N = 9$); p-αKO, 40.5 days ($N = 12$). ****$p < 0.0001$ for KPC vs. αKO; ***$p = 0.0009$ for αKO vs. p-αKO; ****$p < 0.0001$ for KPC vs. p-αKO (log-rank test). (**D**) Percentage of B6 mice that are tumor free at 8 weeks after implantation with KPC (0%), αKO (100%), and p-αKO (0%) cells.

The online version of this article includes the following source data and figure supplement(s) for figure 3:

**Source data 1.** Original western blot files for *Figure 3A*.

**Source data 2.** Mice survival record containing data for *Figure 3C*.

**Figure supplement 1.** In vitro characterization of p-αKO cells.

**Figure supplement 2.** Gene set enrichment analysis (GSEA) of cytokine-related pathways in αKO cells ($n = 3$) vs. p-αKO cells ($n = 3$).

In standard 2D culture, p-αKO cells grew at a similar rate compared to αKO cells (*Figure 3—figure supplement 1A*). Note that both αKO and p-αKO cell lines exhibited significantly slower growth rate than KPC cells. This reduced growth rate explains, at least in part, the longer survival time of p-αKO implanted mice compared with KPC-implanted mice, where the median survival of KPC was 15.5 days and the median survival of p-αKO 40.5 days (*Figure 3C*). Immunoblotting showed no changes in phosphorylation of PIK3CA's downstream effector AKT at both Thr308 and Ser473

residues. As expected, phosphorylation of PRAS40, which is a direct effector of AKT, was also down-regulated in both cell lines lacking PIK3CA as compared to KPC cells. PTEN is a negative regulator of the PI3K–AKT pathway, and its expression was not different between the three cell lines (*Figure 3A*). In summary, the loss of PCCB did not reverse the changes in PIK3CA–AKT signaling in αKO cells. These results indicate that progression of p-αKO tumors is through a pathway independent of PI3K and AKT. In addition, the mitochondrial function evaluated by measuring the oxygen consumption rate (OCR) with Seahorse, as well as the intracellular glycolysis and TCA metabolite changes measured by untargeted liquid chromatography–mass spectrometry (LC–MS) both showed comparable levels between αKO and p-αKO cell lines, indicating that the increased growth of p-αKO cells in vivo is probably not due to PCCB-related mitochondrial metabolic changes (*Figure 3—figure supplement 1B, E*).

## T cells infiltrate p-αKO tumors with increased expression of immune checkpoints

Our previous study demonstrated that regression of αKO tumors is due to T-cell infiltration and subsequent elimination by the host immune system (*Sivaram et al., 2019*). We performed immunohistochemistry (IHC) to assess the status of infiltrating T cells in p-αKO, αKO, and KPC tumors. After the implantation, IVIS imaging was used to confirm the presence of tumors, which were then harvested for analysis. Hematoxylin and eosin (H&E) staining confirmed tumor formation (*Figure 4A*) and immediate adjacent tissue sections were stained for infiltrating T cells using anti-CD3, CD4, and CD8 antibodies. Unexpectedly, p-αKO tumors were also highly infiltrated with T cells, similar to αKO tumors, and as expected, KPC tumors were mostly devoid of T cells (*Figure 4A, B*). Therefore, the conversion of cold KPC tumor into inflamed αKO tumor upon genetic deletion of *Pik3ca* in KPC was not reversed by ablation of PCCB.

As high expression of checkpoint inhibitory receptors is associated with reduced ability to perform cytotoxic functions of CD8+ T cells, we examined whether T-cell checkpoint inhibitory receptors, including programmed cell death 1 (PD-1), cytotoxic T-lymphocyte-associated protein 4 (CTLA-4), T-cell immunoreceptor with Ig and ITIM domains (TIGIT), lymphocyte activating 3 (LAG3), and hepatitis A virus cellular receptor 2 (TIM3) were altered upon PCCB ablation. Tumor tissues harvested from mice implanted with αKO and p-αKO were analyzed by IHC cells positive for the stained markers were quantified (*Figure 4C*). In contrast to αKO tumor sections, p-αKO tumors displayed markedly elevated cell counts positive for PD1, CTLA4, and LAG3. However, despite comparable cell counts, the expression levels of TIGIT and TIM3 appeared stronger in p-αKO tumor sections (*Figure 4D*). These results suggest that infiltrating CD8+ T cells in p-αKO tumors might be functionally exhausted with compromised anti-tumor cytotoxic activity.

To explore the potential involvement of other immunosuppressive cells, flow cytometry analysis was conducted on myeloid cells isolated from αKO and p-αKO tumors. The results revealed a significant twofold increase in M2 macrophage infiltration within the p-αKO tumors. This finding provides additional support for the presence of an immunosuppressive TME specifically within the p-αKO tumors, indicating the complexity of the immune landscape in pancreatic cancer (*Figure 4—figure supplement 1A, B*).

## Inhibition of PD1/PD-L1 checkpoint leads to elimination of most p-αKO tumors

Although multiple immune checkpoints are upregulated in p-αKO tumors, we investigated whether blocking the PD1/PD-L1 immune checkpoint alone would reactivate some of the anti-tumor T cells. We implanted p-αKO cells into a PD1-null mouse line bred in the B6 genetic background (*Figure 5A*). After 8 weeks when the experiment ended, eight out nine mice were still alive and five of the animals were tumor free (*Figure 5B, C*). We next tested if pharmacological blockade of PD1/PD-L1 would achieve a similar result. We implanted p-αKO cells in wild-type B6 mice and then treated them weekly with a neutralizing anti-PD-1 antibody or vehicle control (*Figure 5A*). After 8 weeks of treatment, 9 out 13 anti-PD1-treated mice were alive whereas only 2 out 9 vehicle-treated mice survived (*Figure 5B*). Four mice in the anti-PD1-treated group remained tumor free. In contrast, all vehicle-treated mice developed large pancreatic tumors including the two surviving mice at week #8 (*Figure 5C*).

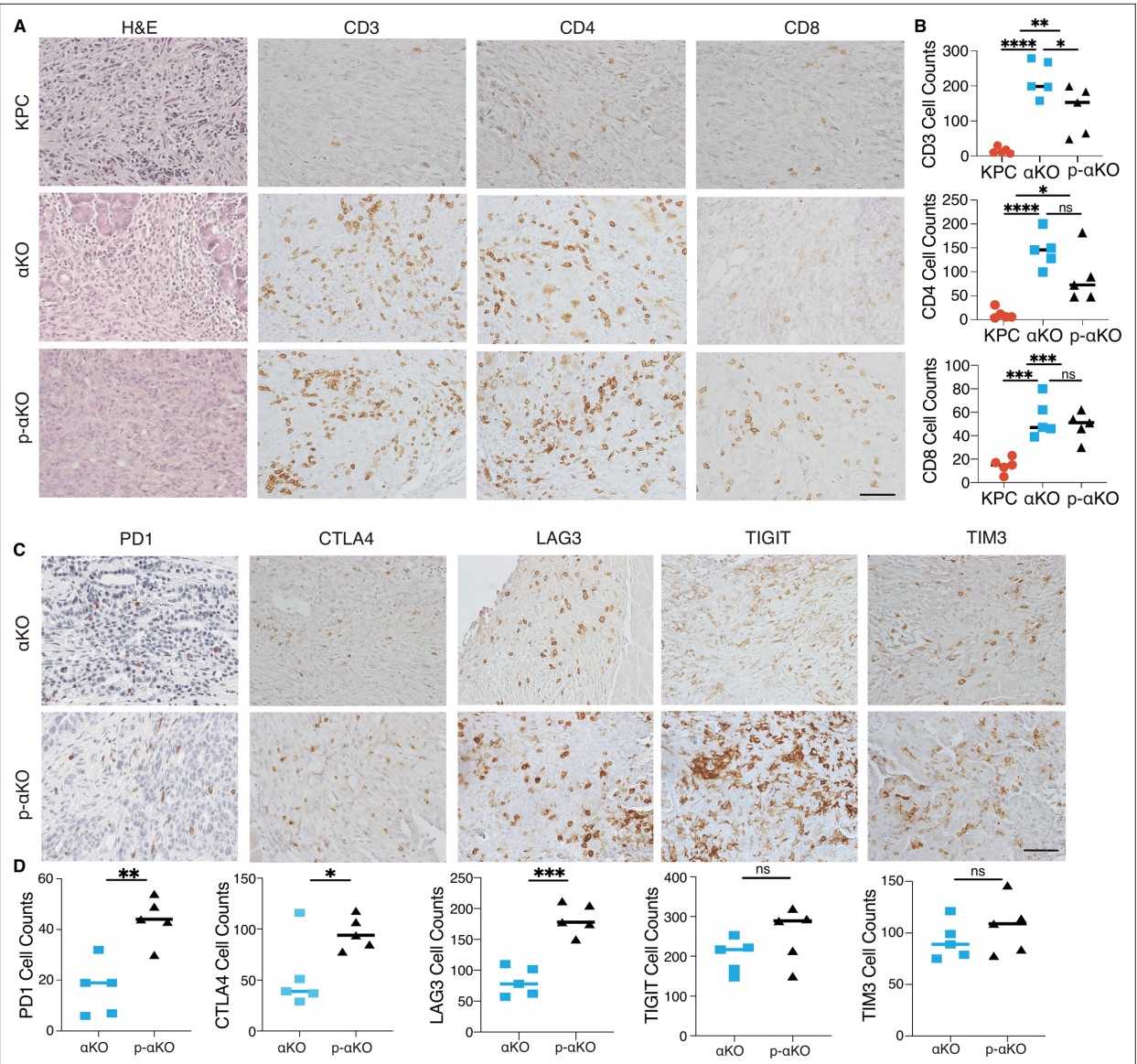

**Figure 4.** T cells infiltrate p-αKO tumors with increased expression of immune checkpoints. (**A**) Pancreatic tissue sections prepared from C57BL/6J mice implanted with KPC, αKO, and p-αKO cells. Tumor sections were stained with hematoxylin and eosin (H&E) and immunohistochemistry (IHC) with CD3, CD4, and CD8 antibodies. Representative sections are shown. Scale bar: 60 μm. (**B**) Quantification of tumor-infiltrating T cells in five representative tumor sections for each group (mean ± SD, n = 3). CD3: ****p < 0.0001 for KPC vs. αKO; **p = 0.0063 for KPC vs. p-αKO; *p = 0.0471 for αKO vs. p-αKO. CD4: ****p < 0.0001 for KPC vs. αKO; *p = 0.0164 for KPC vs. p-αKO; ns for αKO vs. p-αKO. CD8: ***p = 0.0009 for KPC vs. αKO; ***p = 0.0005 for KPC vs. p-αKO; and ns for αKO vs. p-αKO (two-tailed t-test). ns: not significant. (**C**) αKO and p-αKO cells were implanted in the head of pancreas of C57BL/6J mice. Tumor sections were stained with checkpoint markers: PD1, CTLA4, LAG3, TIGIT, and TIM3. Representative sections are shown. Scale bars: 60 μm. (**D**) Quantifications of cells positive for each checkpoint marker at representative tumor sections (mean ± SD, N = 5). αKO vs. p-αKO: **p = 0.0023 for PD1; *p = 0.0371 for CTLA4; ***p = 0.0002 for LAG3; ns for TIGIT and TIM3.

The online version of this article includes the following source data and figure supplement(s) for figure 4:

**Source data 1.** Immunohistochemistry (IHC) counts data for CD3, CD4, CD8, and for PD1, CTLA4, LAG3, TIGIT, and TIM3 for *Figure 4B, D*.

**Figure supplement 1.** Flow cytometry of myeloid cells infiltrated in αKO vs. p-αKO.

## Simultaneous analysis of anti-tumor cytotoxic T cell infiltrating p-αKO tumors by scRNA-seq and TCR repertoire sequencing

To investigate if p-αKO tumors, like αKO cells, also induced clonal enrichment of cytotoxic T cells, we performed a scRNA-seq analysis (same protocol as *Figure 1*) on CD8+ T cells isolated from p-αKO

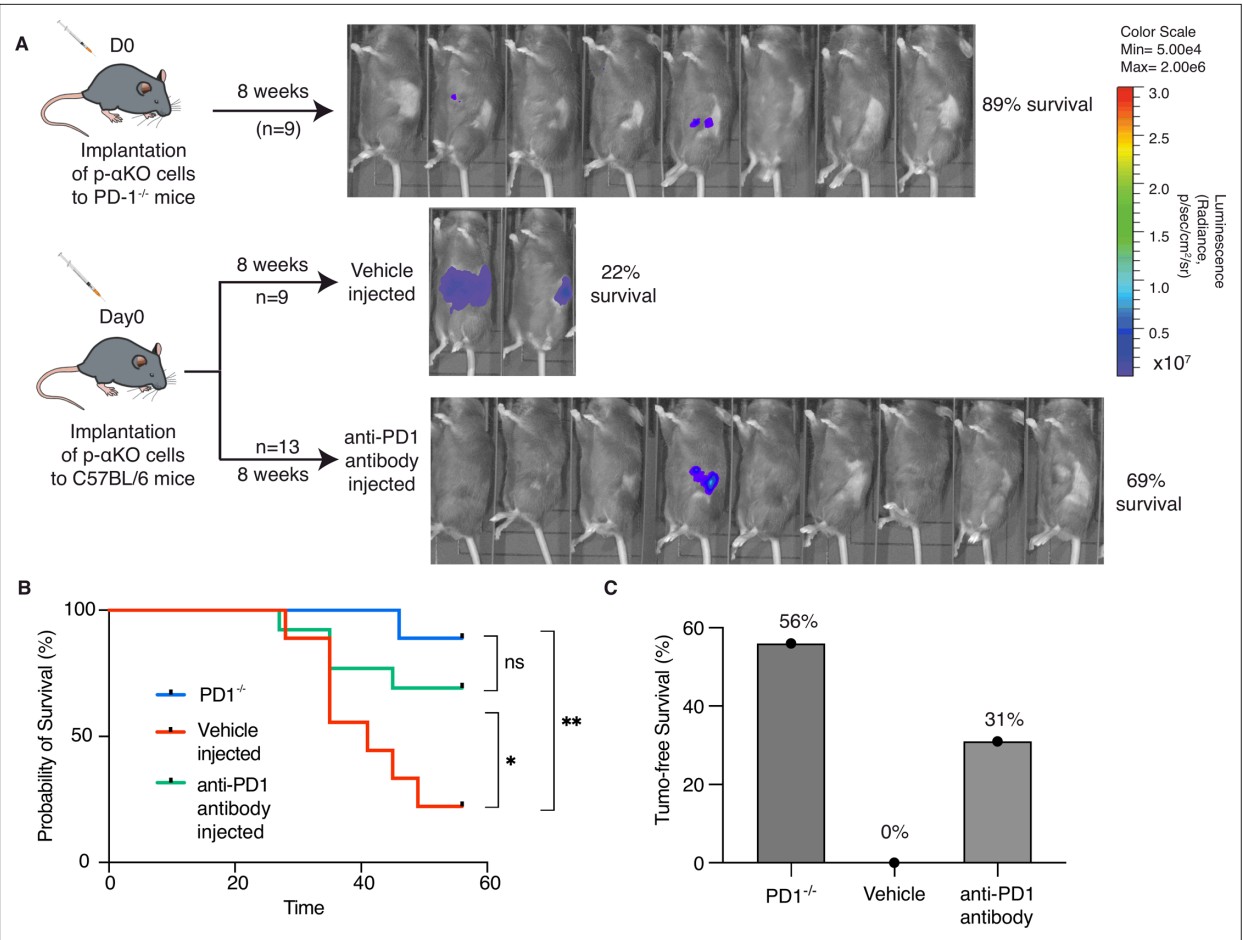

**Figure 5.** Inhibition of PD-1/PD-L1 checkpoint interaction leads to elimination of most p-αKO tumors. (**A**) PD1$^{-/-}$ mice ($N$ = 9) were implanted with p-αKO cells in the pancreas and were monitored for 8 weeks. IVIS imaging was performed to assess tumor size at the end of the experiment. C57BL/6J mice ($N$ = 22) were implanted with p-αKO cells in the pancreas and randomly divided into two groups. One group was injected with 12.5 mg/mg of PD1 neutralizing antibody weekly for 8 weeks, and the vehicle group was injected with the inVivo Pure dilution buffer. IVIS imaging was performed to assess tumor size at the end of the experiment. (**B**) Kaplan–Meier survival curves for PD1$^{-/-}$ mice implanted with p-αKO cells, and vehicle- and anti-PD1-treated C57BL/6J mice implanted with p-αKO cells. Survival rate for PD1$^{-/-}$ group is 89% and median survival undefined; survival rate for vehicle injected group is 22% and median survival is 41 days; survival rate for anti-PD1 injected group is 69% and median survival undefined. *p = 0.0436 for vehicle vs. anti-PD1 antibody injection. **p = 0.003 for vehicle vs. PD1$^{-/-}$ group. (**C**) Tumor-free survival percentage of PD1$^{-/-}$ (56%), vehicle injected (0%), and anti-PD1 antibody injected (31%) groups.

The online version of this article includes the following source data for figure 5:

**Source data 1.** Mice survival record containing data for *Figure 5B*.

tumors implanted in WT mice. In addition, to assess if PD1/PD-L1 blockade reactivated anti-tumor T cells, we also performed the same analysis on CD8$^+$ T cells isolated from p-αKO tumors implanted in PD1-null mice ($n$ = 2 for each group). Similar to what we observed in αKO tumors (see *Figure 1*), there is large clonal enrichment of CD8$^+$ T cells in p-αKO tumors implanted in either WT or PD1-null mice (*Figure 6A*).

The TCR sequences of the large expansions (≥50 cells per clonotype) in both groups are shown in *Figure 6B*. Interestingly, the number of clonotypes with large expansions is much higher in tumors implanted in PD1-null mice. Indeed, the number of clonotypes observed in p-αKO tumors implanted in PD1-null mice is even greater than those found in αKO tumors implanted in WT mice.

Transcriptomic data from all clonal CD8$^+$ T cells were then clustered in an unsupervised fashion and visualized by UMAP (see methods section) and annotated the clusters by ProjecTILs to identify effector memory T cells (EffectorMem), naive-like (or central memory) T cells (Tn), exhausted T cells (Tex), and precursor exhausted T-cell (Tpex) groups (*Figure 6C*). Surprisingly, we observed large

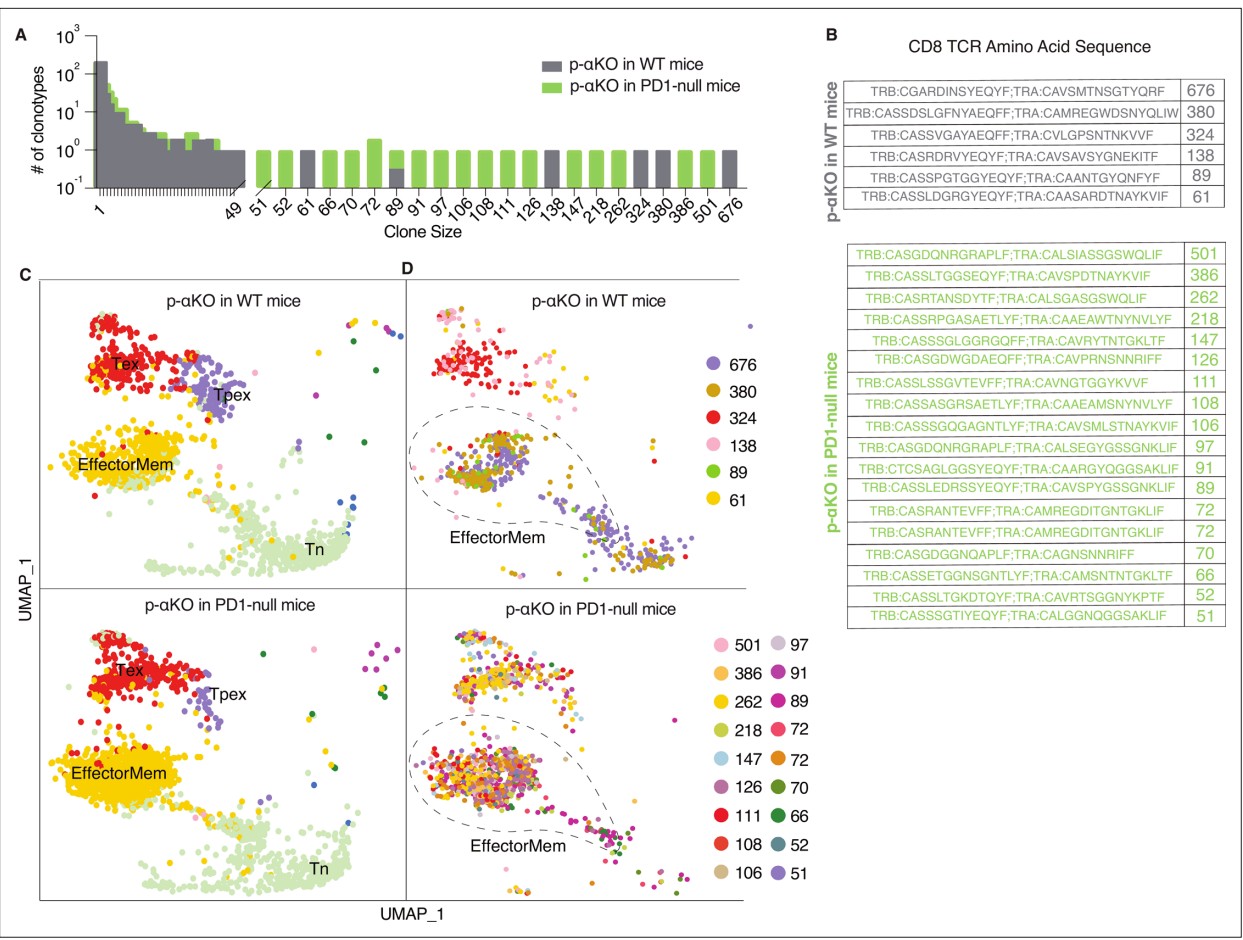

**Figure 6.** Increased number of anti-tumor cytotoxic T cells infiltrating p-αKO tumors implanted in PD1-null vs. WT mice. WT and PD1-null mice in B6 genetic background were implanted with p-αKO cells in the pancreas (n = 2 for each group). Tumors were harvested from these animals 12 days post-implantation. T-cell suspensions prepared from these tumors were subjected to single-cell RNA sequencing (scRNA-seq) and T-cell receptor (TCR) sequencing analysis using the same protocol as **Figure 1**. (**A**) The bar graph shows size distribution of CD8+ T-cell clonotypes found in WT vs. PD1-null tumors. (**B**) CDR3 sequences of clonotypes with size ≥50 (defined as large expansion). Tumors in WT mice had six clonotypes with size ≥50 cells and tumors in PD1-null mice had 18 clonotypes with size ≥50 cells. (**C**) Uniform manifold approximation and projection (UMAP) showing gene expression profiles of all CD8+ T clonotypes found in both groups. EffectorMem: effector memory; Tex: exhausted; Tpex: precursor exhausted; Tn: naive-like or central memory. Each dot corresponds to a single cell. (**D**) UMAP showing gene expression profiles of large expansion CD8+ T-cell clonotypes. Each clonotype is color coded showing the functional mapping of individual cells. The dash line marks where effector memory T cells are located on the UMAP.

The online version of this article includes the following source data for figure 6:

**Source data 1.** Clone size counts for **Figure 6A**.

**Source data 2.** T-cell receptor (TCR) clonotype CDR3 sequences and counts from p-αKO tumors implanted in B6 and PD1KO mice.

number of clonal CD8+ EffectorMem and Tex cells in p-αKO tumors implanted in either WT or PD1-null mice. We next mapped clonotypes with large clonal enrichment (≥50 cells) to the transcriptomic data and displayed the results by clonotype (**Figure 6D**). Again, in both groups, large expansion clonotypes mainly mapped to EffectorMem and Tex gene expression profiles. There are many more CD8+ clonotypes that mapped to the EffectorMem in tumors implanted in PD1-null mice as compared to WT mice. Notably, 856 CD8+ T cells (all 18 large expansion clonotypes) were observed in the EffectorMem group for tumors implanted in PD1-null mice, whereas only 361 cells (from 3 clones) were observed appeared in tumors implanted in WT mice. Indeed, we found more CD8+ EffectorMem cells in p-αKO tumors implanted in PD1-null mice then αKO tumors implanted in WT mice (856 vs. 559). Taken together, these results suggest that regression of p-αKO tumors implanted in mice with PD1/PD-L1 blockade is due to reactivation of anti-tumor cytotoxic T cells.

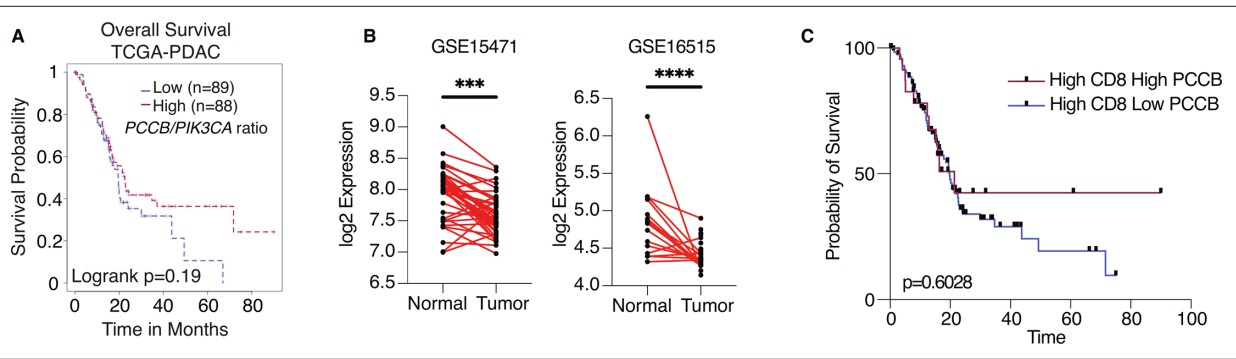

**Figure 7.** Propionyl-CoA carboxylase subunit B (PCCB) regulates pancreatic ductal adenocarcinoma (PDAC) survival in a TIL-dependent manner. (**A**) Overall survival between patients with high *PCCB/PIK3CA* ratio vs. patients with low *PCCB/PIK3CA* of 177 PDAC patients from TCGA database. p = 0.19. (**B**) Two PDAC clinical data GSE15471 and GSE16515 illustrating *PCCB* mRNA expression in tumor sections vs. adjacent normal tissues. ***p = 0.0001 for GSE15471, N = 36 for tumor, and N = 36 for normal; and ****p < 0.0001 for GSE16515, n = 36 for tumor, and N = 16 for normal. (**C**) The overall survival comparisons among PDAC patients with high infiltrating CD8 T-cell level with high or low PCCB expression in TCGA dataset analyzed with TIDE algorithm. High vs. low *PCCB* expression was defined as 80% percentile.

Recent studies propose that immune checkpoint blockade, such as anti-PD1, promotes Tpex to generate effector T cells (*Kallies et al., 2020*; *Im et al., 2016*; *Utzschneider et al., 2016*). Our findings are consistent with this hypothesis, p-αKO tumors implanted in WT mice exhibit a distinct population of Tpex. Treatment with an anti-PD1 antibody resulted in their transformation into effector memory T cells, as observed in p-αKO tumors implanted in the PD1-null mice.

## PCCB expression in human pancreatic cancer

To explore the potential significance of *PCCB* expression in PDAC patients, we queried the Cancer Genome Altas Program (TCGA) database and compared the overall survival of patients with high *PCCB/PIK3CA* ratio to patients with low *PCCB/PIK3CA* ratio. Compared with patients with lower *PCCB/PIK3CA* ratio, patients with higher ratio trended to have better survival (*Figure 7A*). Additional clinical data of PDAC patients were obtained from gene expression omnibus (GEO) database. The GSE15471 is an expression analysis of 36 PDAC patients from the Clinical Institute Funding, where tumors and matching normal pancreatic tissue samples were obtained at the time of surgery. The GSE16515 is the expression data from Mayo Clinic, and it consists of 36 tumor samples and 16 normal samples. Both data revealed significantly lower *PCCB* expressions in tumor sections compared with adjacent normal tissues (*Figure 7B*), which is consistent with our observation that low *PCCB* levels lead to tumor progression. Among the PDAC patients with high CD8+ infiltrating T cells (TCGA PDAC database), patients with high PCCB showed survival benefits over patients with low PCCB expression, which again indicates the T-cell-mediated survival benefits of PCCB expression in PDAC patients (*Figure 7C*).

## Discussion

The PI3K signaling pathway plays an important role in regulating the immune microenvironment in PDAC, and previous studies have indicated that specific PIK3CA inhibitors hold promise in transforming immune-deficient cold tumors into inflamed, hot tumors (*Sun et al., 2021*; *Peng et al., 2016*; *Liu and Sun, 2021*; *Borcoman et al., 2019*; *Sai et al., 2017*; *Sun and Meng, 2020*). Our previous investigations with a PDAC mouse model revealed that downregulation of PIK3CA in tumor cells enhances their recognition and elimination by T cells (*Sivaram et al., 2019*). Building on this foundation, our present study demonstrates that implanting αKO cells into B6 mice induces a clonal enrichment of cytotoxic T cells within the TME.

Conducting a genome-wide in vivo CRISPR screen with αKO cells implanted in the mouse pancreas allowed us to identify key molecules governing the activity of these anti-tumor T cells. Notably, we discovered that the loss of PCCB, a subunit of the mitochondrial enzyme propionyl-CoA carboxylase, can entirely reverse the immune recognition phenotype resulting from PIK3CA downregulation. Deleting PCCB in pancreatic tumor cells lacking PIK3CA leads to immune evasion, tumor progression,

and ultimately, the death of the host animal. Despite T cells still infiltrating p-αKO tumor sections, there is an enrichment of immune checkpoint molecules such as PD1. Intriguingly, implanting p-αKO cells into PD1-null mice significantly improved mouse survival, hinting at the partial attribution of immune evasion in p-αKO tumors to elevated PD1 expression on T cells. Furthermore, treatment of p-αKO tumors with an anti-PD1 neutralizing antibody yielded comparable results, highlighting the potential applicability of this approach in PDAC treatment, especially when coupled with a PIK3CA inhibitor. Noteworthy outcomes ensued when interfering with the PD-L1/PD1 axis, resulting in T-cell-mediated elimination of p-αKO tumor cells. Single-cell sequencing substantiates these findings by confirming that blocking the PD-L1/PD1 interaction reactivates clonal anti-tumor T cells.

Recent investigations underscore the compelling potential of combination therapy involving both PI3K and PD1 inhibitors in cancer treatment. In a study by *Isoyama et al., 2021*, improved anti-tumor effects were observed in melanoma and fibrosarcoma models upon the combination of PI3K inhibitors with PD-1 blockade. This therapeutic strategy raised the levels of tumor-specific CD8[+] T cells thus led to enhanced anti-tumor activity (*Isoyama et al., 2021*). Similarly, *Collins et al., 2022* conducted an in vivo expression screen with a pool of known oncogenes and identified PIK3CA as a factor promoting resistance to anti-PD1 treatment in a colon carcinoma model. Notably, the resistance to immunotherapy was successfully overturned with the administration of a PI3K inhibitor. These findings offer further support to the potential benefits of combining PI3K inhibitors with PD1 blockade for cancer treatment (*Collins et al., 2022*). To further validate the efficacy of this drug combination, future studies can explore its application using our orthotopic implantation model and in genetically modified mouse models that spontaneously develop PDAC. This approach ensures a comprehensive understanding of the therapeutic potential in a context that closely mirrors the intricacies of PDAC progression and the TME.

Although the administration of anti-PD1 neutralizing antibody yielded promising results, the treatment alone did not achieve 100% efficacy. Indeed, our scRNA-seq result from PD1-null mice revealed that not all exhausted T cells were reactivated, potentially explaining why some of the mice still succumb to tumor progression despite PD-L1-/PD1 blockade. In addition, to improve survival rates, it is crucial to incorporate other immunotherapies targeting additional checkpoint inhibitory receptors, such as CTLA4, LAG3, and TIGIT. While anti-CTLA4 and anti-PD1 are standard treatments for many solid tumors, TIGIT and LAG3 have emerged as promising new targets for cancer immunotherapy. LAG3, although demonstrating limited efficacy as a monotherapy, is frequently used in combination with other checkpoint inhibitors (*Gulhati et al., 2023*). Similarly, while anti-TIGIT antibodies have shown remarkable effectiveness in treating solid tumors, the dual blockade of PD-1 and TIGIT has demonstrated superior clinical benefits in cancer treatment (*Smith et al., 2013*; *Tedelind et al., 2007*; *Høgh et al., 2020*). Our IHC data notably revealed a significant upregulation of both TIGIT and LAG3 in p-αKO tumors, suggesting that the inclusion of additional immune checkpoint inhibitors, alongside PD1 inhibition, may further improve treatment outcomes. Therefore, future studies that evaluate blocking other immune checkpoints such as TIGIT, LAG3, CTLA4, or TIM3, in combination with anti-PD1, could provide valuable insights.

While our findings highlight the involvement of PCCB in regulating T-cell activities, the precise underlying mechanism remains unclear. Previous studies primarily focused on the role of *Pccb* in the genetic disease propionic acidemia, caused by mutations in *Pccb* or *Pcca* subunits, which impair PCC function (*Wongkittichote et al., 2017*). Dysfunctional PCC leads to the accumulation of propionyl-CoA and other byproducts such as propionylcarnitine (C3), methylcitrate (MC), and 3-hydroxypropionate (3OHPA) (*Kurczynski et al., 1989*; *Ando et al., 1972*). Intracellular metabolite analysis by LC–MS showed elevated MC and C3 levels upon *Pccb* deletion, although the statistical significance was not reach (*Figure 3—figure supplement 1D*). Importantly, intracellular propionyl-CoA levels were similar between αKO and p-αKO cells, potentially due to its conversion to MC and C3. However, the precision of propionyl-CoA detection remains challenging due to its limited quantity. Additional intracellular metabolites assessed through non-targeted LC–MS exhibited noteworthy alterations, suggesting their possible involvement in the regulation of PD1 expression on T cells (*Figure 3—figure supplement 1C*). However, further experiments are required to obtain a better understanding of how PCCB-regulated metabolic pathways affect T-cell function.

Various studies have highlighted the anti-inflammatory properties of propionate, a byproduct of dysfunctional PCC. *Smith et al., 2013* demonstrated that propionic acid regulates colonic Treg (cTreg)

homeostasis in the gut lumen. Propionic acid supplementation in the drinking water of germ-free (GF) mice increased cTreg frequency and number in the gut lumen, while Tregs in the spleen, mesenteric lymph nodes, and thymus remained unaffected. Treatment of cTregs isolated from GF mice with propionate led to increased Foxp3 and IL10 expressions, both of which play important roles in Treg-mediated immune suppression. Additionally, cTregs exhibited a higher proliferation rate when co-cultured with propionate (*Smith et al., 2013*). Another study by *Tedelind et al., 2007* reported that propionate suppressed the secretion of TNFα and IL8 from neutrophils, inhibited the NF-κB pathway, and downregulated immune-related genes in a colon cell line (*Tedelind et al., 2007*). Recently, Hogh et al. reported that propionate upregulated the surface expression of the immune stimulatory NKG2D ligands MHC-class-I-related family (MICA/B) on several cancer cells and activated T cells, mediated by mTORC2 (*Høgh et al., 2020*). These indirect observations suggest a link between *Pccb* and immune responses although the published results are somewhat conflicting.

Additionally, cytokines and other mechanisms within the TME may contribute to the upregulation of PD1 expression on T cells in p-αKO tumors. RNA sequencing of αKO and p-αKO cells revealed substantial changes in several cytokine-related signaling pathways, indicating potential cytokine involvement in regulating PD1 expression on T cells (*Figure 3—figure supplement 2*). Moreover, comparing cells grown in vitro to tumor growth in vivo poses challenges due to the complex TME and potential crosstalk between immune cells and the metabolic system. Future studies should conduct single-cell RNAseq analysis of in vivo tumor cells present within the TME to uncover the mechanisms of immune evasion upon deletion of *Pccb*.

Our study is the first to report an association between PCCB and PIK3CA. We utilized PIK3CA knockout cells (αKO) instead of conventional PIK3CA inhibitors to avoid off-target effects reported in previous studies (*Wang et al., 2017*; *Gharbi et al., 2007*; *Miller et al., 2019*; *Kong and Yamori, 2008*), providing a more specific and accurate assessment of PIK3CA's role in regulating immune responses in PDAC. We employed an orthotopic mouse model for our CRISPR screen, which more closely mimics the TME of human PDAC. Our genome-wide screen study identified PCCB as a key regulator of anti-tumor T-cell activity. We showed that PCCB regulates immune evasion by promoting increased PD-L1 expression in anti-tumor cytotoxic T cells. Consequently, our findings suggest that the resistance of PDAC to PI3K inhibitors and checkpoint inhibitors could potentially be surmounted by deploying both drugs in combination. This may be a promising avenue for future pre-clinical and clinical studies to directly assess if combining these FDA-approved drugs, possibly in conjunction with other treatment modalities, will improve outcomes for pancreatic cancer patients.

## Methods
### Cell culture and reagents
KPC, αKO, and p-αKO were all cultured in complete DMEM media (Gibco, Thermo Fisher Scientific) with 10% FBS (Corning 35-015-CV) and 1% P/S (penicillin/streptomycin: Gibco 15140-122), incubated at 37°C with 5% $CO_2$. The KPC cell line was a gift from Dr. David Tuveson (Cold Spring Harbor) and was confirmed to possess a G-to-D mutation in the KRAS gene (KRASG12D) and a silent mutation in the transformation related protein 53 gene (TP53R172H) by mutation analysis in our previous study (*Sivaram et al., 2019*). The preparation of KPC and αKO cell lines were described before (*Sivaram et al., 2019*). Briefly, KPC cells were infected with lentiviral particles containing CMV-firefly luciferase with a neomycin selection marker (Cellomics Technology, PLV-10064). After 48-hr transfection, KPC cells were selected with 1.5 mg/ml G418, and the luciferase expression was confirmed by the IVIS Lumina III imaging system (Xenogen). KPC cells that express luciferase signals are referred to as WT KPC. *Pik3ca*$^{-/-}$ (αKO) KPC cell lines were generated by transfecting WT KPC cells with *Pik3ca* CRISPR/Cas9 αKO and HDR plasmids (Santa Cruz Biotechnology, sc-422231 and sc-422231-HDR). Transfected cells were selected with 5 mg/ml puromycin and sorted based on red fluorescent protein (RFP) by fluorescence-activated cell sorting on a FACSAria (BD Biosciences). RFP$^+$ cells were serially diluted in a 96-well plate to generate single-cell clones, and clones that showed an absence of PIK3CA protein on western blot were retained (referred to as αKO cell lines). In order to knock out *Pccb* from αKO cell lines, αKO was transfected with *Pccb* CRISPR/Cas9 KO plasmid (Santa Cruz Biotechnology, sc-426258) and incubated for 48 hr. Again, western blot was used to verify PCCB expression and cells without the presence of PCCB and PIK3CA are referred to as p-αKO cell lines. The sgRNA oligonucleotides

used for knockout Pik3ca are sc-422231 A: GCGCACTATTTATGACCCAG; sc-422231 B: TCACCATG CCGTCATACTCC; sc-422231 C: CAGAAGTCCAAGACTTTCGA. The sgRNA oligonucleotides used for knockout *Pccb* are sc-426258 A: GAGTCATTGAGCCCGATCAC; sc-426258 B: CAGATGTGCCGA CTTCGGAA; sc-426258 C: ACTGGACGGGGCCGAATCAA.

## GeCKO v2 lentiviral library preparation and infection

The mouse GeCKO v2 library was obtained from Addgene (#1000000052). LentiCRISPRv2 is a one-vector plasmid system for the mouse GeCKO (Genome-scale CRISPR Knockout) pooled library. The sublibrary A contains 67,405 sgRNAs targeting 20,611 protein-coding genes, 1175 microRNAs and 1000 control sgRNAs. To ensure no loss of representation, the GeCKO library A was amplified by first electroporating the library to Endura ElectroCompetent cells and then transforming to LB agar plates. The colonies from the LB agar plates were harvested and purified by Maxiprep. To produce lentivirus, the plasmid was used to transfect HEK293FT cells by lipofectamine 2000 plus reagent with packaging plasmids pVSVg and psPAX2. After incubating for 2 days, the lentiviral supernatant from the HEK293FT cells was pooled and filtered to get rid of cellular debris. Then the lentiviral titer was determined through transduction. To calculate the Multiplicity Of Infection (MOI) and optimal kill curve, cells were treated with a serial dilution of concentrated virus followed by puromycin selections the next day with titrations of 0, 0.5, 1, and 2 mg/ml. After 48 hr of treatment, cell apoptosis was measured with PI and Hoechst staining to determine the concentrations of viral titer and puromycin that give approximately an MOI of 0.2–0.4 with nearly 100% killing (1:10 dilution). To ensure 350× of library coverage, $1 \times 10^7$ αKO cells were seeded, and infected with 1:10 dilutions of the viral titer aiming for an MOI of 0.2–0.4.

## CRISPR knockout screen in an orthotopic pancreatic implantation mouse model

After incubating the transduced αKO cells for 2 days, $7.5 \times 10^5$ of the transfected cells in 30 ml of PBS were prepared for each mouse and were implanted in the pancreas of nine immunocompetent wild-type C57BL/6J mice (B6; Stock#000664). To monitor tumor progression, 100 mg/kg RediJect D-Luciferin (PerkinElmer 770504) was injected intraperitoneally into mice followed by imaging on the IVIS Lumina III imaging system (Xenogen) to monitor tumor progression. Data were analyzed using Living Image v4.3.1 software.

## NGS and data analysis

Genomic DNA was extracted both from the primary tumor and metastatic sites to perform NGS in order to identify enriched sgRNAs. QIAGEN DNeasy Blood & Tissue Kit (#13323) was used to extract DNA according to the manufacturer's protocol. The concentration of DNA was determined by Nano-Drop ND 1000. NGS libraries were prepared by a two-step PCR described before (*Chen et al., 2015*; *Joung et al., 2017*). All PCR reactions were performed with Herculase II Fusion DNA Polymerase (Agilent K3467). The first PCR (PCR1) utilized primers specific to the sgRNA-expression vector to amplify the sgRNA containing region, in order to preserve full complexity of the GeCKO library. To achieve a 350-fold coverage of the sgRNA library, approximately 200 mg of gDNA for each sample was used, and thus approximate 67 PCR1 reactions were performed for each sample (3 µg gDNA/50 µl reaction). The second PCR (PCR2) was performed to add barcoded adaptors—P7 and P5—to the products from the first PCR, so it allowed multiplex sequencing on Illumina NextSeq. For our 67,405 mGeCKOa sgRNA library, at least 7 PCR2 reactions were performed (one 100 µl reaction per $10^4$ constructs in the library). The primers used for first round PCR are as follows: PCR1-F: CCCGAGGG GACCCAGAGAG; PCR1-R: GCGCACCGTGGGCTTGTAC. The primers used for the second round PCRs are as follows: Fwd: AATGATACGGCGACCACCGAGATCTACACTCTTTCCCTACACGACGCT CTTCCGATCT-stagger-barcodes-TCTTGTGGAAAGGACGAAACACCG; Rev: CAAGCAGAAGAC GGCATAC-GAGAT-barcode-GTGACTGGAGTTCAGACGTGTGCTCTTCCGATCT-stagger-TCTACTAT TC-TTTCCCCTGCACTGT. The barcode used for tumor 1 is AAGTAGAG; and the barcode used for tumor 2 is CGCGCGGT. Before sequencing, DNA quality and quantity were determined by Qubit dsDNA BR Assay Kit and Agilent 2200 TapeStation Analysis. All samples were sequenced on Illumina NextSeq 550 sequencer. The sequencing data were processed for sgRNA representation. The 8 bp barcodes were first demultiplexed from the sequencing reads, followed by adapter trimming. The

remaining spacer sequences were aligned to the designed sgRNA library by bowtie, with tolerance of a single-nucleotide mismatch. The number of mapped sequences was imported into R/RStudio to quantify the total number of reads. Besides, the computational tool MAGeCK was used to confirm the identified mapping counts.

## Orthotopic pancreatic implantation mouse model

All the C57BL/6J mice (#000664) and PD-1$^{-/-}$ mice (#028276) were purchased from Jackson Laboratory. The orthotopic implantation surgeries have been described previously (*Sivaram et al., 2019*). Briefly, cells were trypsinized and washed twice with PBS, then counted with a cell counter to prepare $5 \times 10^5$ cells in 30 µl of PBS per mouse for injection (or $7.5 \times 10^5$ cells for the screen). Mice were anesthetized with a combination of 100 mg/kg ketamine and 10 mg/kg xylazine, followed by a small vertical incision made over the left lateral abdominal area. The pancreas was then located with the aid of a light microscope, and the injection was made at the head of pancreas by a sterile Hamilton syringe with a 27-gauge needle. After sutures, mice were given an intraperitoneal injection of 2 mg/kg ketorolac.

## Mouse survival studies

Mice implanted with tumor cells (KPC, GeCKO-treated αKO, αKO cells, and p-αKO cells) were monitored by IVIS imaging weekly. Mice with weight loss >15% body weight and/or inability to move were euthanized as these were considered the endpoint. For mice implanted with p-αKO cells, week 8 was set as the endpoint of the experiment and any surviving animals were euthanized.

## Single-cell isolation

Pancreas with tumor tissues were harvested from mice on day 12 after tumor implantation. After excluding lymph nodes, tissues were washed with cold PBS and minced with scissors in the hood. The minced tissues were washed with PBS, then digested in 5 ml of 1 mg/ml Collagenase type V (Worthington LS005282) solution dissolved in HBSS for 20 min at 37°C. 2 ml of Roche Red Blood Cell Lysis Buffer (Millipore Sigma 11814389001) was used per sample to get rid of red blood cells, then passed through 40 µm cell strainer to generate single cells.

## Enrichment of CD4$^+$ and CD8$^+$ T cells from dissociated tissues

10 µl of CD4 (TIL) MicroBeads (Miltenyi Biotec 130-116-475) and 10 µl of CD8 (TIL) MicroBeads (Miltenyi Biotec 130-116-478) were added to 90 µl of single-cell suspensions (up to 10 million cells each) and incubated at 4°C for 15 min. After adding the depletion wash buffer to a final volume of 500 µl for up to 50 million cells, the mixture was separated in an LS column (Miltenyi Biotec 130-042-401) by the magnetic field of the MACS Quadro separator. The flow-through were CD4- and CD8-negative cells, and the magnetically labeled cells that were flushed out after removing the magnetic field were CD4- or CD8-positive T cells. The isolated cells were centrifuged at 300 × *g* 4°C for 5 min and resuspended in PBS with 0.1% of BSA to generate 1e6/ml of cells.

## scRNA-seq library preparation with 10× Genomics platform

We followed the manufacturer's protocol provided by 10× Genomics for the preparation of Chromium NextGEM Single Cell 5′ gene expression libraries. Single-cell suspensions at a concentration of 0.7–1.2e6/ml were mixed with reverse transcription reagents, Gel Beads containing barcoded oligonucleotides, and partitioning oil on a microfluidic chip to form Gel Beads in emulsion (GEMs). Within each GEM, a single cell was lysed and the mRNAs barcoded and reverse transcribed to cDNA, followed by PCR amplification of the barcoded cDNAs. scRNA-seq and VDJ libraries were prepared using the Chromium Next GEM Single Cell 5′ Kit v2 (PN-1000265) and the Chromium Single Cell Mouse TCR Amplification kit (PN-1000254) at the Northport VAMC Single Cell Facility. The gene expression and VDJ libraries were pair-end sequenced (PE150) through a commercial supplier (Novogene, Inc) at a depth of 20,000 and 5000 reads/cell, respectively.

## scRNA-seq data analysis

The Cell Ranger v7.0.0 pipeline was used to demultiplex and align sequencing data to the 'ENSEMBL GRCm39' mouse transcriptome to generate gene expression matrices. The Cell Ranger VDJ pipeline

was used to assemble sequences and identify paired clonotypes on the V(D)J libraries. The gene expression matrices were further analyzed by the Seurat R package. Three quality control criteria were employed to filter the matrices to exclude unwanted sources of variations: number of detected transcripts, genes, and percentage of reads mapping to mitochondrial genes. Cells with unique molecular identifiers (UMIs) over the interquartile range of 95% (potential doublets) and under 1000 (potential fragments) or a mitochondrial proportion higher than 20% (potential apoptotic) were removed. Moreover, we used the Doublet Finder R algorithm to further eliminate doublet contamination. After the quality control, we annotated cellular identity using the R package SingleR, which assigns each cell to a reference type that has the most similar expression profile with. In further analysis, sctransform package in Seurat was used to normalize UMI count data. The principal component analysis of high-dimensional data was performed to identify highly variable genes in each sample, and the top principal components were selected for unsupervised clustering of cells with a graph-based clustering and visualized with UMAP.

## Western blotting

Cells were washed twice by PBS followed by lysis in RIPA buffer 50 mM HEPES, pH 7.4, 10 mM sodium pyrophosphate, 50 mM NaF, 5 mM EDTA, 1 mM sodium orthovanadate, 0.25% sodium deoxycholate, 1% NP40, 1 mM PMSF, and protease inhibitor cocktail (Sigma P8340). After centrifuge, cell lysates were collected, and proteins were separated by SDS–PAGE gel followed by semi-dry transfer onto nitrocellulose membranes. After blocking with 5% milk in Tris-buffered saline plus 0.1% Tween 20 (TBST), primary antibodies prepared in a 1:1000 dilution with TBST for overnight. After incubating the membrane with horseradish peroxidase-linked secondary antibodies (Thermo Fisher Scientific #62-6520 and #31460) in a 1:5000 dilution with TBST, signals were developed by adding ECL reagent or SuperSignal West Femto (Thermo Scientific 34095). Signals were detected by a FluorChem E imager (ProteinSimple). Primary antibodies used: PCCB: Millipore Sigma HPA036939; PIK3CA: Cell Signaling #4249S; HSP90: Cell Signaling #4874S; PTEN: Cell Signaling #9559S; pAKT (Thr308): Cell Signaling #9275S; pAKT (Ser473): Cell Signaling #4051S; AKT 1/2/3: Santa Cruz Biotechnology sc-377556; actin: Millipore Sigma #A2103; PPRAS40: Cell Signaling #2997S.

## Cell proliferation assay

2500 cells were plated in triplicates for each cell line on a 12-well plate on day 0. From days 1 to 4, cells were washed with PBS, trypsinized and resuspended with completed media and then stained with trypan blue (Invitrogen T10282) with a 1:1 dilution. The cells were then counted using Countess II automated cell counter (Life Technologies). Cell numbers were plotted by GraphPad Prism 9.

## Real-time metabolic analysis

The Seahorse XF96 extracellular flux analyzer (Seahorse Bioscience, USA) was utilized to measure the OCR. Cells were cultured in 96-well XF microplates with 20,000 cells per well and incubated for 18 hr at 37°C. To evaluate mitochondrial activity, sequential injections of oligomycin, the electron transport chain uncoupler FCCP, and specific inhibitor of the mitochondrial respiratory chain were performed. The OCR data were normalized to cell number.

## LC–MS

To assess the total metabolite levels, cells were plated in 6-well plates in triplicates with equal cell numbers for each well. After cells were washed three times with PBS, a mixture of methanol, acetonitrile, and water with formic acid was quickly added to the dishes. The plates were then placed on ice for 5 min, followed by the addition of 15% $NH_4HCO_3$ to neutralize the acetic acid. The cells were collected by scraping the centrifuged, and the resulting supernatant was used for LC–MS analysis. The LC–MS employed hydrophilic interaction chromatography (HILIC) coupled with electrospray ionization to a Q Exactive PLUS hybrid quadrupole orbitrap mass spectrometer (Thermo Scientific). A specific LC separation was conducted using an XBridge BEH Amide column, with a gradient of solvents A and B. The gradient varied over a specific time frame, and the flow rate, injection volume, and column temperature were controlled. The metabolite features were extracted using MAVEN software, considering the labeled isotope and a mass accuracy window. The isotope natural abundance

and tracer isotopic impurity were corrected using AccuCor. The LC–MS and data analysis was done by Metabolomics Shared Resource at the Rutgers Cancer Institute of New Jersey.

## Histology

Mouse tumors were fixed in 10% formalin for 24 hr. Tissue processing was done by Leica ASP300S Tissue Processor. After embedding with paraffin, tissues were sectioned into 4 μm sections. H&E staining was performed using H&E. IHC was performed by deparaffinization and rehydration of the tissue sections, followed by antigen retrieval in citrate buffer, pH 6.0 (Vector Laboratories H-3300-250) using a Decloaking Chamber (Biocare Medical). To block endogenous peroxidase activity, 3% $H_2O_2$ (Fisher Scientific 7722-84-1) was used. And endogenous biotin was blocked by using the Avidin/Biotin Blocking kit (Vector Laboratories SP-2001). After incubation of primary antibodies (1:1000) overnight, the signal was developed by the R.T.U. Vectastain Kit (Vector PK-7100) and 3,3'-diaminobenzidine in chromogen solution (Agilent K3467). Hematoxylin (Agilent CS70030-2) then used to counterstain the nucleus. Antibodies used in IHC were CD3: Abcam #ab16669; CD4: Abcam #ab183685; CD8α: Cell Signaling #98941S; PD1: Abcam #ab214421; CTLA4: Abcam #ab237712; LAG3: Abcam #ab209238; TIGIT: Abcam #ab300073; TIM3: Abcam #ab2413332.

## IHC quantification

Two consecutive sections were obtained from each sample. Each section two or three images with ×40 magnification were taken on an Olympus BX43 with DP26 camera. The number of cells was manually counted by QuPath.

## Anti-PD1 antibody treatment

The InVivoPlus anti-mouse PD1 (CD279) antibody was purchased from BioxCell (#BP0273). The PD1 antibody was diluted in InVivoPure pH7 Dilution buffer (BioxCell #IP0070) to make a 12.5 μg/mg (average of 22 mg mice) solution. After mice were implanted with p-αKO cells, 100 μl of prepared mixture or the dilution buffer were injected intraperitoneally to mice weekly for 8 weeks. Mice were imaged by IVIS from weeks 5 to 8, and mice survival data were recorded for survival studies.

## Statistical analysis

All statistical analyses were done by GraphPad Prism 9. For all statistical comparisons, the two-tailed Student's t-test was used for comparison between two groups. The one-way ANOVA with Bonferroni's post hoc test and Kruskal–Wallis test with Dunn's post hoc test were used for multi-group comparisons. p values less than 0.05 were considered statistically significant.

## Study approval

All experiments in mice were conducted in accordance with the Office of Laboratory Animal Welfare and approved by the IACUC of Stony Brook University, Stony Brook, New York.

## Acknowledgements

We express our sincere appreciation to our dedicated laboratory scientists, Lisa Ballou and Giuseppe Caso, as well as our undergraduate research assistant Mark Koch and Nathaniel Tchangou for their unwavering commitment, expertise, and contributions to this project. This study was supported in part by the Department of Veterans Affairs Merit Review BX004083 (RZL) and the National Institutes of Health R21CA274425 (RZL), R01CA129536 (WXZ), and R01CA236246 (WXZ).The funders had no role in study design, data collection, and interpretation, or the decision to submit the work for publication.

# Additional information

## Funding

| Funder | Grant reference number | Author |
|---|---|---|
| Veterans Affairs Scholarship Program | BX004083 | Richard Z Lin |
| National Institutes of Health | R21CA274425 | Richard Z Lin |
| National Institutes of Health | R01CA129536 | Wei-Xing Zong |
| National Institutes of Health | R01CA236246 | Wei-Xing Zong |

The funders had no role in study design, data collection and interpretation, or the decision to submit the work for publication.

## Author contributions

Han V Han, Conceptualization, Data curation, Software, Formal analysis, Visualization, Methodology, Writing – original draft; Richard Efem, Software, Formal analysis, Visualization, Methodology; Barbara Rosati, Resources, Data curation, Methodology; Kevin Lu, Sara Maimouni, Ya-Ping Jiang, Valeria Montoya, Ando Van Der Velden, Methodology; Wei-Xing Zong, Methodology, Project administration; Richard Z Lin, Conceptualization, Resources, Supervision, Investigation, Project administration, Writing – review and editing

## Author ORCIDs

Han V Han http://orcid.org/0009-0000-0788-3019
Richard Z Lin https://orcid.org/0000-0002-3967-0859

Reviewer #1 (Public Review): https://doi.org/10.7554/eLife.96925.2.sa1
Reviewer #2 (Public Review): https://doi.org/10.7554/eLife.96925.2.sa2
Reviewer #3 (Public Review): https://doi.org/10.7554/eLife.96925.2.sa3
Author response https://doi.org/10.7554/eLife.96925.2.sa4

---

# Additional files

## Supplementary files

Supplementary file 1. T-cell receptor (TCR) clonotype, CDR3 sequences, and counts from two αKO tumors, two p-αKO tumors, and two p-αKO implanted in PD1KO mice tumors.

Supplementary file 2. Sequencing-based counting of sgRNA frequencies with respective gene name from our CRISPR screen for the two tumors named T1 and T2.

Supplementary file 3. Seahorse data measuring oxygen consumption rate (OCR) for KPC, αKO, and p-αKO cells.

Supplementary file 4. Metabolic changes for αKO and p-αKO cells measured by liquid chromatography–mass spectrometry (LC–MS).

MDAR checklist

## Data availability

scRNA-seq data from Figures 1 and 6 have been deposited in GEO under accession code GSE254041. CRISPR Screen data from Figure 2 have been deposited in BioProject under the accession code PRJNA1068774.

The following datasets were generated:

| Author(s) | Year | Dataset title | Dataset URL | Database and Identifier |
|---|---|---|---|---|
| Han H, Lin R | 2024 | Depletion of PCCB and/ or PIK3CA from pancreatic tumor cells KPC influences T cells infiltrating to TME | https://www.ncbi. nlm.nih.gov/geo/ query/acc.cgi?acc= GSE254041 | NCBI Gene Expression Omnibus, GSE254041 |
| Han H, Lin R | 2024 | CRISPR Screen identifies regulators for T cells activities in pancreatic cancer | https://www.ncbi.nlm. nih.gov/bioproject/? term=PRJNA1068774 | NCBI BioProject, PRJNA1068774 |

The following previously published datasets were used:

| Author(s) | Year | Dataset title | Dataset URL | Database and Identifier |
|---|---|---|---|---|
| Badea L, Herlea V, Dima SO, Dumitrascu T | 2009 | Whole-Tissue Gene Expression Study of Pancreatic Ductal Adenocarcinoma | https://www.ncbi. nlm.nih.gov/geo/ query/acc.cgi?acc= GSE15471 | NCBI Gene Expression Omnibus, GSE15471 |
| Pei H, Li L, Fridley BL, Jenkins GD | 2009 | Expression data from Mayo Clinic Pancreatic Tumor and Normal samples | https://www.ncbi. nlm.nih.gov/geo/ query/acc.cgi?acc= GSE16515 | NCBI Gene Expression Omnibus, GSE16515 |

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
