## [Editor Report · eLife assessment]

The significance of the findings is **valuable**, with implications for immunotherapy design in pancreatic ductal adenocarcinoma. The evidence was considered **incomplete** and partially supportive of the major claims.

---

## [Referee Report · Reviewer #1 (Public Review)]

Summary:

Pancreatic ductal adenocarcinoma (PDAC) is an aggressive disease that does not respond to immunotherapy. This work represents an extension of the authors' prior observation that PI3Ka deletion in an orthotopic KPC pancreatic tumor model confers susceptibility to immune-mediated elimination. The authors' major claims in the present manuscript are as follows:

(1) PI3Ka (Pik3ca) knockout in KPC pancreatic tumor cells induces clonal T cell expansion.

(2) Genome-wide LOF screen in aKPC cells to identify tumor-intrinsic determinants of PI3Ka-KO-enhanced T cell response identified Pccb.

(3) When Pccb is knocked out in the context of Pi3ka knockout KPC, anti-tumor T cell response is reduced as measured by

a. Increased tumor progression

b. Decreased survival

c. T cells are still clonally expanded but less functional

(4) ICB is able to "reactivate" clonally expanded T cells.

(5) Conclusion: Pccb modulates the activity of T cells in PDAC.

Overall, the experiments were appropriately executed and technically sound, albeit underpowered for single-cell analyses. Upon careful consideration of the data, the biggest weakness of the paper is the authors' interpretations of results, particularly for claims 1 and 4 (see below for details). Much of the data is correlative and does not delve into causation, leaving this reviewer wishing for experiments that would clearly demonstrate that Pccb in tumor cells directly impacts T cell anti-tumor activity.

Strengths:

(1) Tumor intrinsic determinants of intratumoral T cell infiltration in PDAC are less commonly evaluated as combination therapies for ICB. This is a point of conceptual innovation and importance.

(2) A sensitized CRISPR screen to identify mutations that rescue KPC/PI3Ka-KO tumors from immune-mediated killing is an elegant method to better understand the molecular mechanisms contributing to KPC immunosurveillance. Further, one screen candidate (Pccb) was experimentally validated.

(3) Single-cell clonotype analyses hold promise for identifying tumor-reactive T cells (though authors never demonstrated that specific clones were tumor antigen specific).

Weaknesses:

(1) "Clonal expansion of cytotoxic T cells infiltrating the pancreatic αKO tumors"

a. Only two tumor-bearing hosts were evaluated by single-cell TCR sequencing, thus limiting conclusions that may be drawn regarding repertoire diversity and expansion.

b. High abundance clones in the TME do not necessarily have tumor specificity, nor are they necessarily clonally expanded. They may be clones which are tissue-resident or highly chemokine-responsive and accumulate in larger numbers independent of clonal expansion. Please consider softening language to clonal enrichment or refer to clone size as clonal abundance throughout the paper.

c. The whole story would be greatly strengthened by cytotoxicity assays of abundant TCR clones to show tumor antigen specificity.

(2) "A genome-wide CRISPR gene-deletion screen to identify molecules contributing to Pik3ca-mediated pancreatic tumor immune evasion"

a. CRISPR mutagenesis yielded outgrowth of only 2/8 tumors. A more complete screen with an increased total number of tumors would yield much stronger gene candidates with better statistical power. It is unsurprising that candidates were observed in only one of the two tumors. Nevertheless, the authors moved forward successfully with Pccb.

(3) T cells infiltrate p-αKO tumors with increased expression of immune checkpoints

a. In Figure 4D, cell counts are not normalized to totalCD8+ T cell counts making it difficult to directly compare aKO to p-aKO tumors. Based on quantifications from Figure 4D, I suspect normalization will strengthen the conclusion that CD8+ infiltrate is more exhausted in p-aKO tumors.

b. Flow cytometric analysis to further characterize the myeloid compartment is incomplete (single replicate) and does not strengthen the argument that p-aKO TME is more immunosuppressive.

c. It could, however, strengthen the argument that TIL has less anti-tumor potential if effector molecule expression in CD8+ infiltrating cells were quantified.

(4) Inhibition of PD1/PD-L1 checkpoint leads to elimination of most p-αKO tumors

a. It is reasonable to conclude that p-aKO tumors are responsive to immune checkpoint blockade. However, there is no data presented to support the statement that checkpoint blockade reactivates an existing anti-tumor CD8+ T cell response and does not instead induce a de novo response.

b. The discussion of these data implies that anti-PD-1 would not improve aKO tumor control, but these data are not included. As such, it is difficult to compare the therapeutic response in aKO versus p-aKO. Further, these data are at best an indirect comparison of the T cell responsiveness against tumor, as the only direct comparison is infiltrating cell count in Figure 4 and there are no public TCR clones with confirmed anti-tumor specificity to follow in the aKO versus p-aKO response.

---

## [Referee Report · Reviewer #2 (Public Review)]

Summary:

Pancreatic ductal adenocarcinoma is generally considered a "cold" tumor type with little T cell infiltration. This group demonstrated previously that deletion of the PIK3CA isoform of PI3K in the orthotopic pancreatic ductal adenocarcinoma KPC mouse tumor model led to the elimination of tumors by T cells. Here they performed a genome-wide gene-deletion screen in this tumor using CRISPR to determine what was required for this T cell-mediated infiltration and tumor rejection. Deletion of Pccb in the tumors, which encodes propionyl-CoA carboxylase subunit B, allowed for the outgrowth of the PIK3CA-deleted KPC tumors. This was confirmed with the specific deletion of Pccb in the tumor cells. Demonstrating a likely role in tumor progression in human patients as well, high expression of PCCB in pancreatic ductal adenocarcinoma correlated with lower patient survival. T cells still infiltrated these tumors, but had much higher expression of exhaustion markers. Blockade of PD-1 signaling allowed for the rejection of these tumors. While these are intriguing data demonstrating that loss of PCCB by pancreatic ductal adenocarcinoma is a mechanism to escape T cell immunity, the mechanism by which this occurs is not determined. In addition, there are a few issues that suggest the conclusions of the manuscript should be tempered.

Strengths:

In vivo analysis of tumor CRISPR deletion screen.

The study describes a possible novel mechanism by which a tumor maintains a "cold" microenvironment.

Weaknesses:

(1) A major issue is that it seems these data are based on the use of a single tumor cell clone with PIK3CA deleted. Therefore, there could be other changes in this clone in addition to the deletion of PIK3CA that could contribute to the phenotype.

(2) The conclusion that the change in the PCCB-deficient tumor cell line is unrelated to mitochondrial metabolic changes may be incorrect based on the data provided. While it is true that in the experiments performed, there was no statistically significant change in the oxygen consumption rate or metabolite levels, this could be due to experimental error. There is a trend in the OCR being higher in the PCCB-deficient cells, although due to a high standard deviation, the change is not statistically significant. There is also a trend for there being more aKG in this cell line, but because there were only 3 samples per cell line, there is no statistically significant difference.

(3) More data are required to make the authors' conclusion that there are myeloid changes in the PCCB-deficient tumor cells. There is only flow data from shown from one tumor of each type.

(4) The previous published study demonstrated increased MHC and CD80 expression in the PIK3CA-deficient tumors and these differences were suggested to be the reason the tumors were rejected. However, no data concerning the levels of these proteins were provided in the current manuscript.

---

## [Referee Report · Reviewer #3 (Public Review)]

Summary:

In this study, Han and co-authors showed that implantation of Pik3ca deficient KPC cells (aKO) induced clonal expansion of CD8 T cells in the tumor microenvironment. Using aKO cells, they conducted an in vivo genome-wide gene-deletion screen, which showed that deletion of propionyl-CoA carboxylase subunit B gene (Pccb) in αKO cells (p-aKO) leads to immune evasion and tumor progression. Eventually, mice injected with p-aKO but not aKO succumbed to their tumors. Similar to the parental aKO cell line, p-aKO tumors were still infiltrated with clonally expanded CD8+ and CD4+ T cells, as shown by the IHC. Further analyses showed that T cells infiltrating p-aKO tumors expressed high levels of exhaustion markers (PD-1, CTLA-4, TIM3, and TIGIT). Furthermore, PD-1 signaling blockade using PD-1 mAb or genetic depletion of PD-1 reactivated the infiltrated T cells, controlling tumor progression and improving the overall mice survival. Thus, the authors concluded in the abstract that "Pccb can modulate the activity of cytotoxic T cells infiltrating some pancreatic cancers." Although the data clearly showed that the loss of Pccb facilitated the immune evasion of pancreatic cancer cells, there is no clear evidence provided that Pccb deletion can actually modulate the activity of CD8 T cells. One may argue that the deletion of Pccb reduces the immunogenicity of the p-aKO cancer cells, making them less susceptible to killing by normally functional CD8+ T cells.

Strengths:

In vivo, Crisper-Cas-9 screen using tumor cell lines.

Identify a gene that could reduce the immunogenicity of cancer cells.

Weaknesses:

The IHC technique that was used to stain and characterize the exhaustion status of the tumor-infiltrating T cells.

---

## [Author Response]

We appreciate the reviewers' detailed feedback, which has highlighted several areas where our study could be strengthened. Although we acknowledge the relatively limited scope of our CRISPR-based gene-deletion screen, we successfully demonstrated the immunogenic role of Pccb in our syngenetic pancreatic cancer mouse model. Specifically, loss of PCCB in our mutant KRAS/p53 PIK3CA-null (αKO) cells blocked host T cell killing of tumor cells.

Furthermore, blocking the PD1/PD-L1 interaction reverses this anti-tumor immunogenic effect. We agree with the reviewers regarding the limitations of our study, such as the sample size in our scTCR sequencing and the lack of direct cytotoxicity assays to confirm tumor-specific T cell clones. However, our results are consistent across multiple experimental approaches that strongly suggest meaningful differences in host T cell response to the three implanted tumor types, KPC, αKO and p-αKO. We agree that future mechanistic studies will be important to determine how PCCB is involved in this immunogenic response. We also agree with the reviewers that future additional studies with other KPC cell lines will strength our conclusion regarding PCCB. Finally, we acknowledge the inherent limitations of IHC techniques to assess the involvement of other T cell checkpoints that might also be involved in this anti-tumor immunogenic effect. In summary, despite these limitations, our findings provide novel insight into the role of PCCB in pancreatic tumor immunogenicity and contribute to the ongoing discussion of how to improve therapeutic strategies for this deadly cancer.

**Reviewer 1:**
Weaknesses:(1) Clonal expansion of cytotoxic T cells infiltrating the pancreatic αKO tumorsa. Only two tumor-bearing hosts were evaluated by single-cell TCR sequencing, thus limiting conclusions that may be drawn regarding repertoire diversity and expansion.

We agree with the reviewer that possible repertoire diversity and expansion could be observed by sequencing more tumor-bearing hosts. However, our current data reveal a marked consistency in the transcriptional expression within the two tumors analyzed per group. Importantly, these features are significantly divergent between the αKO and p-αKO groups. While recognizing the limited sample size, the observed within-group consistency and the clear distinction between groups strongly support the validity of the reported trends.

b. High abundance clones in the TME do not necessarily have tumor specificity, nor are they necessarily clonally expanded. They may be clones which are tissue-resident or highly chemokine-responsive and accumulate in larger numbers independent of clonal expansion. Please consider softening language to clonal enrichment or refer to clone size as clonal abundance throughout the paper.

We agree with the reviewer that it’s possible that the high abundance clones are not necessarily tumor specific. Our previous work (N. Sivaram 2019) demonstrated the critical role of increased pancreatic CD8+ T cells in αKO tumor regression within B6 mice. Therefore, antigen specific CD8+ T cell clonal expansion within the pancreas is an anticipated observation. However, as the reviewer pointed out, a portion of this expansion may be attributable to factors independent of tumor antigens. While the low T cell infiltration observed in KPC-implanted mice argues against a purely tissue-resident explanation, further investigation is required to definitively establish the tumor specificity of individual clones. We have revised the manuscript to reflect this nuance, replacing "clonal expansion" with "clonal enrichment".

c. The whole story would be greatly strengthened by cytotoxicity assays of abundant TCR clones to show tumor antigen specificity.

As mentioned above, we agree with the reviewer that future studies are needed to investigate each of the specific clones. Due to the extended timeframe required, it’s beyond the scope of the present study.

(2) A genome-wide CRISPR gene-deletion screen to identify molecules contributing to Pik3camediated pancreatic tumor immune evasion"a. CRISPR mutagenesis yielded outgrowth of only 2/8 tumors. A more complete screen with an increased total number of tumors would yield much stronger gene candidates with better statistical power. It is unsurprising that candidates were observed in only one of the two tumors. Nevertheless, the authors moved forward successfully with Pccb.

We agree that by including more mice in the CRISPR screen, it’s possible that we could have identified more candidates. Regardless, we have successfully demonstrated PCCB’s role in pancreatic tumorgenicity with our mouse model.

(3) T cells infiltrate p-αKO tumors with increased expression of immune checkpoint

*a. In Figure 4D, cell counts are not normalized to totalCD8+ T cell counts making it difficult to directly compare aKO to p-aKO tumors. Based on quantifications from Figure 4D, I suspect normalization will strengthen the conclusion that CD8+ infiltrate is more exhausted in p-aKO tumors. *

Due to the use of distinct tumor sections for quantifying CD8+ cells and T cell checkpoint inhibitory receptor expression, direct normalization of these counts is challenging. However, we observed comparable CD8+ cell numbers between αKO and p-αKO tumors, with p-αKO tumors exhibiting nearly double the expression of immune checkpoint receptors. Therefore, even accounting for potential normalization discrepancies, we anticipate that p-αKO tumors would still demonstrate a significantly higher percentage of immune checkpoint receptorpositive cells compared to αKO tumors.

b. Flow cytometric analysis to further characterize the myeloid compartment is incomplete (single replicate) and does not strengthen the argument that p-aKO TME is more immunosuppressive. It could, however, strengthen the argument that TIL has less anti-tumor potential if effector molecule expression in CD8+ infiltrating cells were quantified.

We agree that including more tumor samples will strengthen the argument that p-αKO TME is more immunosuppressive. Future studies need to be done to characterize CD8+ T cells.

(4) Inhibition of PD1/PD-L1 checkpoint leads to elimination of most p-αKO tumorsa. It is reasonable to conclude that p-aKO tumors are responsive to immune checkpoint blockade. However, there is no data presented to support the statement that checkpoint blockade reactivates an existing anti-tumor CD8+ T cell response and does not induce a de novo response

We agree that future studies exploring the clonotypes of T cells infiltrating tumors in PD-1treated mice are necessary to determine whether observed T cell response represents reactivation of existing clones, a de novo response, or a combination of both.

b. The discussion of these data implies that anti-PD-1 would not improve aKO tumor control, but these data are not included. As such, it is difficult to compare the therapeutic response in aKO versus p-aKO. Further, these data are at best an indirect comparison of the T cell responsiveness against tumor, as the only direct comparison is infiltrating cell count in Figure 4 and there are no public TCR clones with confirmed anti-tumor specificity to follow in the aKO versus p-aKO response.

Since αKO tumors completely regress with 100% animal survival, we deemed anti-PD1 treatment in this group unnecessary. While we did assess anti-PD1 treatment in KPCimplanted mice, no survival benefit was observed (data not shown). The p-αKO tumor model was the only one in which anti-PD1 treatment improved survival. The complexity of the in vivo tumor microenvironment likely contributes to the lack of shared TCR clones between αKO and p-αKO tumors, even within the same tumor group. Future studies aimed at identifying tumorspecific clones may involve transferring in vivo models to in vitro assays or the generation of novel mouse strains expressing identified TCRs. However, these approaches require substantial time and resources and are beyond the scope of the present study.

**Reviewer 2:**
Weaknesses:(1) A major issue is that it seems these data are based on the use of a single tumor cell clone with PIK3CA deleted. Therefore, there could be other changes in this clone in addition to the deletion of PIK3CA that could contribute to the phenotype.

We have previously tested a different KPC cell line (DT10022) with genetically downregulated PIK3CA and found mice implanted with αKO cells also showed tumor regression. However, we have not tested if deletion of Pccb in the DT10022-aKO cell line will have the same effect.

1. The conclusion that the change in the PCCB-deficient tumor cell line is unrelated to mitochondrial metabolic changes may be incorrect based on the data provided. While it is true that in the experiments performed, there was no statistically significant change in the oxygen consumption rate or metabolite levels, this could be due to experimental error. There is a trend in the OCR being higher in the PCCB-deficient cells, although due to a high standard deviation, the change is not statistically significant. There is also a trend for there being more aKG in this cell line, but because there were only 3 samples per cell line, there is no statistically significant difference.

Although PCCB is known to cause metabolic changes, in the context of this study, we are comparing PCCB-deficient to PCCB & PIK3CA double-deficient cells. We did not address if PCCB loss alone would cause metabolic alteration. We suspect that is the case.

(3) More data are required to make the authors' conclusion that there are myeloid changes in the PCCB-deficient tumor cells. There is only flow data from shown from one tumor of each type.

We agree that including more tumor samples will strengthen the argument that p-αKO TME is more immunosuppressive.

(4) The previous published study demonstrated increased MHC and CD80 expression in the PIK3CA-deficient tumors and these differences were suggested to be the reason the tumors were rejected. However, no data concerning the levels of these proteins were provided in the current manuscript.

Our previous hypothesis for altered MHC and CD80 levels is based on the observation that there is a dramatic increase in the number of infiltrating T cells upon Pik3ca deletion. In this study, similar levels of infiltrating T cells were observed when Pccb was deleted in αKO cells, therefore we do not expect any changes in MHC and CD80 levels since these tumors appears to be still recognized by the T cells. Indeed, we are able detect clonal enrichment in p-αKO tumors.

**Reviewer 3:**
Weaknesses:The IHC technique that was used to stain and characterize the exhaustion status of the tumorinfiltrating T cells.

We agree with the reviewer that incorporating multi-color IHC or flow cytometry to characterize the exhaustion status of specific T cell subtypes would provide more comprehensive information. Unfortunately, we do not have the resources to perform these studies currently.